



# Spring Blooms in the Baltic Sea have weakened but lengthened from 2000 to 2014

Philipp M. M. Groetsch[1, 2], Stefan G. H. Simis[3, 4], Marieke A. Eleveld[1], and Steef W. M. Peters[2]

[1]Institute for Environmental Studies (IVM). De Boelelaan 1087, 1081 HV Amsterdam, The Netherlands
[2]Water Insight. Marijkeweg 22, 6709 PG Wageningen, The Netherlands
[3]Plymouth Marine Laboratory. Prospect Place, The Hoe, PL1 3DH Plymouth, United Kingdom
[4]Finnish Environment Institute SYKE. Erik Palménin Aukio 1, 00560 Helsinki, Finland

*Correspondence to:* Philipp M. M. Groetsch (groetsch@waterinsight.nl)

**Abstract.** Phytoplankton spring bloom phenology was derived from a 15-year time-series (2000-2014) of ship-of-opportunity chlorophyll-*a* fluorescence observations in the Baltic Sea. Decadal trends were analysed against inter-annual variability in bloom timing and intensity, and environmental drivers (nutrient concentration, temperature, radiation level, wind speed).

Spring blooms developed along a south to north gradient with the first blooms peaking mid-March in the Bay of Mecklenburg and the latest peaks occurring mid-April in the Gulf of Finland. Bloom duration was similar between sea areas ($43 \pm 2$ d), except for shorter bloom duration in the Bay of Mecklenburg ($36 \pm 11$ d). Variability in bloom timing increased towards the south. Bloom peak concentrations were highest (and most variable) in the Gulf of Finland ($20.2 \pm 5.7$ mg m$^{-3}$) and the

Bay of Mecklenburg ($12.3 \pm 5.2$ mg m$^{-3}$).

    Bloom peak concentration showed a negative trend of $-0.31 \pm 0.10$ mg m$^{-3}$ yr$^{-1}$. Trend-agnostic distribution-based (Weibull-type) bloom metrics showed a positive trend in bloom duration of $1.04 \pm 0.20$ d yr$^{-1}$, which was not found for any of the threshold-based metrics. The Weibull bloom metric results were considered representative in presence of bloom intensity trends.

Bloom intensity was mainly determined by winter nutrient concentration, while bloom timing and duration co-varied with meteorological conditions. Longer blooms corresponded to higher water temperature, more intense solar radiation, and lower wind speed. It is concluded that nutrient reduction efforts led to decreasing bloom intensity, while changes in Baltic Sea environmental conditions associated with global change correspond to a lengthening spring bloom period.

## 1 Introduction

Human influence and climate change transform terrestrial and marine ecosystems worldwide at unprecedented rates (Cleland et al., 2007; Cloern et al., 2015). Coastal marine systems experience anthropogenic pressure as well as indirect changes in climatic conditions, which affect the marine



food-web (Heisler et al., 2008; Zhai et al., 2013; Paerl and Huisman, 2008). Ecosystem responses

to these changes are difficult to relate to unique causes (HELCOM, 2007b; Winder and Cloern, 2010; Neumann et al., 2012). Experiments designed to support biogeochemical model scenarios (e.g. Neumann et al., 2002; Tamminen and Andersen, 2007; Seppälä and Olli, 2008) help to disentangle observed trends. However, the predictive capabilities of biogeochemical models (e.g Kuusisto et al., 1998; Meier et al., 2011; Gnanadesikan and Anderson, 2009) remain dependent on calibration

against long and consistent multi-variable time series.

Phytoplankton bloom intensity and timing (bloom phenology) are indicators for ecosystem health at the base of the food web (e.g. Hays et al., 2005; Adrian et al., 2009; Vargas et al., 2009). Phenological studies are increasingly used to inspect regional ecosystem response to nutrient reduction efforts (HELCOM, 2007a; Voss et al., 2011; Fleming-Lehtinen et al., 2015) and changing climatic

conditions (Sommer and Lengfellner, 2008; Paerl and Huisman, 2009). The Baltic Sea is a coastal ecosystem affected by eutrophication (Korpinen et al., 2012), which intensifies naturally occurring spring- and summer bloom (Bianchi and Engelhaupt, 2000; HELCOM, 2007a). The Helsinki Commission formulated a nutrient reduction scheme aimed at improving ecosystem health in 1992 (HELCOM, 2008), which entered into force in 2000. Monitoring of key ecosystem health indicators is

implemented in the national monitoring programmes of HELCOM contracting parties. These programmes include traditional dedicated sampling campaigns at sea and increasingly the use of highly resolving observation platforms.

Ships-of-opportunity (typically cargo ships or passenger ferries) offer a largely weather-independent, reliable, and cost-effective platform for the collection of high frequency in situ ob-

servations (Leppänen et al., 1995; Ainsworth, 2008). Phytoplankton pigment fluorometers are included in most so called ferryboxes. In the Baltic sea, such systems have recorded phytoplankton bloom on the route from Helsinki to Travemünde (v.v.) since 1992 (Rantajärvi et al., 2003). On this route ferryboxes have collected over 9.5 million chlorophyll-$a$ pigment fluorescence observations from 1926 transects with a median revisit time of under two days in the last 15 years (2000-2014).

Ship-based observations from merchant vessels provide continuity, which is particularly important in seasons when other observation systems are less reliable. In spring, satellite observations are rare due to high average cloud cover, while high costs of dedicated research cruises and coastal laboratories limit their spatio temporal coverage. Ferrybox observations are therefore the primary source of observations to study spring bloom dynamics in this region.

Phytoplankton abundance and succession in the Baltic Sea is controlled by nutrient (Neumann et al., 2002; Tamminen and Andersen, 2007) and light availability (Sverdrup, 1953; Smetacek and Passow, 1990; Nelson and Smith, 1991; Siegel et al., 2002), mixing-status (Ueyama and Monger, 2005; Sharples et al., 2006), temperature (Grayek and Staneva, 2011), ice cover (Kahru and Nommann, 1990; Omstedt et al., 2004; Sommer and Lengfellner, 2008), and salinity (Fennel, 1999;

Tamminen and Andersen, 2007). In addition, the quantum yield of fluorescence is influenced by so-



lar irradiance (Kiefer, 1973; Dandonneau and Neveux, 1997; Marra, 1997; Sackmann et al., 2008), species composition, and physiology (Kiefer et al., 1989). Hence, interpretation of unattended pigment fluorescence measurements in terms of phytoplankton biomass presents a number of challenges (Roesler and Barnard, 2013). Firstly, phytoplankton distribution exhibits high spatial and temporal

variability, while ferryboxes measure pigment fluorescence at fixed depth (Ruokanen et al., 2003). Therefore, stratified conditions may not be well represented in the data (Groetsch et al., 2014). Secondly, in a typical ferrybox setup fluorescence yield is at best determined as daily sea area-average, which disregards variability on smaller spatio-temporal scales. Despite these challenges, Fleming and Kaitala (2006) demonstrated that ferrybox observations in the Baltic Sea can be used to derive

bloom timing and intensity for biomass-rich sea areas. The authors reported a slightly negative trend in bloom initiation in the Northern Baltic Proper and the Gulf of Finland for the period 1992-2004. Recent studies also reported shifts in phytoplankton spring bloom biomass or species composition (e.g. Klais et al., 2011; Wasmund et al., 2011, 2013), but shifts in Baltic Sea spring bloom timing are still lacking.

Choosing an adequate bloom metric is not trivial as no strict guidelines exist that unambiguously recommend one metric over the other. Bloom metrics for both remotely sensed and in situ sampled time series are commonly divided into three groups: 1) fixed or variable concentration threshold metrics (Siegel et al., 2002; Fleming and Kaitala, 2006; Lips et al., 2014; Racault et al., 2015), 2) growth-rate-based metrics (Rolinski et al., 2007; Wiltshire et al., 2008), and 3) distribution-based

metrics (Rolinski et al., 2007; Platt et al., 2009; Vargas et al., 2009; Zhai et al., 2011). Threshold- and growth-rate based metrics typically require data pre-processing (e.g. interpolation and smoothing), to mitigate the impact of gaps, noise, outliers, and multi-modal bloom distributions to the derived bloom phenology (Rolinski et al., 2007; Cole et al., 2012; Ferreira et al., 2014). Distribution-based metrics fit an analytical expression to observations using fitting routines designed to cope with im-

perfections in the input data while optimally preserving natural variability. Distribution-based bloom metrics are considered more robust than threshold- or growth-rate-based metrics, in the presence of complex, multi-modal bloom observations (Ji et al., 2010). Interpretation based on several, conceptually different bloom metrics can be used to obtain uncertainty estimates (Ho and Michalak, 2015), and to qualitatively indicate long-term trends in bloom phenology. The latter is because threshold-

based metrics are biased by long-term bloom intensity trends, whereas growth-rate and distribution-based metrics are not. Figure 1 illustrates how a gradual decline (negative trend) in bloom peak concentration causes any threshold-based metric to introduce an artificial negative trend in bloom duration. In contrast, growth-rate and distribution-based metrics yield a constant bloom duration for the given example because they are sensitive to concentration distributions, rather than absolute

concentrations.

The aims of this study are twofold: (1) to report long-term trends for Baltic Sea spring bloom intensity and timing, and (2) to attribute these trends to changes in environmental conditions. This



paper describes a methodology to derive quality controlled time-series of chlorophyll-*a* concentrations from observations collected by the Baltic Sea Alg@line program and its predecessors over
a period of 15 years (2000-2014). Uncertainties arising from variability in the phytoplankton pigment fluorescence yield are estimated. Bloom phenology parameters based on several conceptually differing bloom metrics are presented, and explored for long-term trends. Inter-annual variability of bloom phenology parameters are attributed to nutrient availability and meteorological conditions (temperature, radiation level, wind speed), which might help to relate long-term trends to unique causes. Finally, we summarize how these results contribute to the discussion on recent changes in the Baltic Sea, and the monitoring practices that need to be in place to detect such changes.

## 2 Materials and Methods

### 2.1 Alg@line Data

In situ data in this study were collected until 2009 by the Finnish Institute of Marine Research, and by the Finnish Environment Institute (SYKE) from 2009 onwards, within the Alg@line network of Baltic Sea ferryboxes. Here we consider systems installed on two cargo vessels, M/S *Finnpartner* (2000-2006) and M/S *Finnmaid* (2007-2014), which served between Travemünde (Germany) and Helsinki (Finland) as depicted in Fig. 2. Three routes were sailed during the study period. Depending on wave height and direction the passage between Gotland and the mainland of Sweden (52 %) was favoured over the direct route east of Gotland (39 %), while the route with a lay-over in Gdansk (Poland) was only occasionally served during 2009 to 2012 (7 %). Several transects (2 %) were sailed for refuelling or maintenance in other ports and not used for this study.

Details on the instrumentation of the Alg@line ferrybox systems can be found in Leppänen et al. (1994); Rantajärvi et al. (2003); Ruokanen et al. (2003); Seppälä et al. (2007). In summary, the systems recorded in vivo fluorescence of chlorophyll-*a* (chla), salinity and temperature throughout the studied period (2000-2014). Turbidity and (in summer) phycocyanin pigment fluorescence were recorded from 2005 onwards and are not used here. At cruising speed (20 kn to 23 kn) the sampling interval of 20 s resulted in a nominal spatial resolution of 200 m.

Quality control flags were defined from (1) sensor reading thresholds on speed, flow rate, hull and line temperatures, and (2) data variability, as lower and upper bounds for standard deviation between neighbouring measurements. Measurements at low ($< 5$ kn) or zero ship speed are typically collected in harbour and were disregarded. Erroneous records, e.g. caused by instrument communication errors, were removed using a moving window mean filter. A window length of 25 observations (approximately 8.3 min) was used for records of ship speed, and a window length of 100 observations (33.3 min) was used for flow rate and temperature records. Low flow rates can indicate blocked passages, pump failure, or leaks. Flow meter readings were available for approximately one-third of all records. A proxy for flow disruption is the difference in ship-hull temperature and in-line tem-



perature. Flow rates $< 0.3\,\mathrm{L\,min^{-1}}$ or a temperature difference $> 2\,^{\circ}\mathrm{C}$ were used to flag records as suspect. Instrument failure, communication and digitizing errors may lead to 'stuck' values, which

were detected by calculating standard deviation in a moving window of 100 samples. Observations corresponding to low standard deviation ($\sigma < 1e^{-4}$) of chla fluorescence measurements or GPS-derived latitude were omitted. GPS-derived latitude was additionally filtered for exceptionally high short-term variability ($\sigma > 0.5$, window size 50 samples), caused by poor satellite reception or serial communication errors. Table 1 provides an overview of the applied quality control flags.

Chla fluorescence data were corrected for sensor drift and discontinuities by transect-wise normalization (division by transect mean). This was necessary to account for changes in instrumentation, signal contamination due to bio-fouling, trapped bubbles and particles, and changes in sensor sensitivity due to deterioration or manual adjustments.Laboratory analysis results of bottle samples are typically available from every $6^{th}$ transect, with up to 24 samples collected by automated, re-

frigerated water samplers (Teledyne Isco). Laboratory analyses included inorganic nutrient concentrations (nitrate+nitrite, phosphate and silicate), chla concentration, and occasionally inverted light microscopy counts of phytoplankton species. Laboratory chla concentration results were used to convert transect-normalized chla fluorescence to units of chla concentration (in $\mathrm{mg\,m^{-3}}$). First, a linear (generalized least squares) regression fit of normalized chla fluorescence against corresponding chla

lab measurements was carried out for each sampled transect. If the regression failed ($R^2 < 0.3$ or $p > 1$) a moving window regression was carried out (window length 10 samples) and the subset with the highest $R^2$ was used to determine the correction factor. The threshold for $R^2$ was determined manually based on the distribution of $R^2$, while $p > 1$ indicates numerical instabilities during the fitting procedure. Each transect without corresponding bottle samples was corrected by individually

applying the regression parameters of the two neighbouring sampled transects. These two solutions were then interpolated linearly, weighted by their temporal distance to the respective transect. Negative concentration values occasionally occurred for weak fluorescence signals, and were set to zero.

    The diurnal variability of the fluorescence signal was estimated from quality-controlled observations in all seasons. First, these observations were divided by their respective transect mean to

remove biomass-driven first-order variability in the fluorescence signal. Then, diurnal cycles were derived by dividing these observations into hourly bins and sun elevation angle ranges (0.1 rad bins).

## 2.2 Meteorological Data

Photosynthetically active radiation (PAR), sea surface temperature (SST) and wind speed (WIND) were derived from the ECMWF ERA-Interim reanalysis data set (Dee et al., 2011). The spatial reso-

lution of the model is constrained by the underlying atmospheric model, which is stored on a spatial T255 grid corresponding to approximately 79 km cell size when projected to a reduced Gaussian grid. Four values per day were retrieved for each parameter and the entire Baltic Sea. Parameter values for each Alg@line observation were extracted using spatio-temporal spline interpolation of





third order. The first order seasonal signal (e.g. rising PAR and SST in spring) was removed from
the observations by subtracting multi-year (2000-2014) daily sea area averages, approximated by
second order polynomials. The seasonally detrended parameters were then averaged over the bloom
period and are further referred to as PAR, SST, and WIND.

### 2.3 Nutrient Concentration and Depletion Timing

A single term for nutrient availability was adopted from Fleming and Kaitala (2006), calculated
as NUT $= \sqrt[3]{(NO_3 + NO_2) \times PO_4 \times SiO_4}$, where $NO_3 + NO_2$, $PO_4$ and $SiO_4$ are the concen-
trations of nitrite+nitrate, phosphate, and silicate, respectively. NUT was spatially binned for each
investigated sea area and re-sampled to daily averages and consecutively smoothed with a 21-day
centred-running-mean filter. This treatment resembles the Alg@aline processing (see section 2.4) to
enable consistent interpretation of the joint data set. Nutrient concentrations and depletion timing
are described using the following metrics. The nutrient concentration prior to bloom start (NUT-
PEAKVALUE) was defined as the yearly maximum nutrient concentration (day-of-year between 31
and 160). The day-of-year when the nutrient concentrations equalled 100 %, 50 %, and 25 % of their
peak values are referred as NUT-PEAKDAY, NUT-DEPLDAY-50, and NUT-DEPLDAY-25. The day and
value of the lowest nutrient concentration are referred to as NUT-MINDAY and NUT-MINVALUE. The
rate of nutrient depletion between 75 % and 25 % of the peak value (NUT-SLOPE) was determined
through linear regression.

### 2.4 Extraction of Bloom Timing and Intensity

Extraction of bloom timing and intensity was carried out for five Baltic Sea areas, where each area
follows definitions of the HELCOM Combine program (HELCOM, 2013). Figure 2 illustrates the
location of the areas: the Western Gulf of Finland (GOF: >59.5 °N latitude, along-transect), the
Northern Baltic Proper (NBP: 58.4-59.5 °N latitude, along-transect), the combined Western and East-
ern Gotland basins (GOT: 56.2-58.4 °N latitude, along-transect), the Southern Baltic Proper (SBP:
54.5-56.2 °N latitude, along-transect), and the Bay of Mecklenburg (BOM: <54.5 °N latitude, along-
transect). For the GOT and SBP areas only routes that passed by Gotland were selected whereas routes
via Gdansk were excluded. This is because the route through Gdansk was sailed only from 2009 to
2012. If not otherwise stated, all further steps are carried out individually for each of these areas and
for day-of-year between 31 (1 January) and 160 (9 June).

Alg@line chla concentrations (see section 2.1) were resampled to daily sea area averages, using
linear interpolation, and subsequently smoothed with a 21-day centred-running-mean filter (e.g. Fer-
reira et al., 2014; Racault et al., 2015) to fill in gaps and reduce short-term variability. We derive
several metrics, all of which have in common that the bloom peak concentration (PEAKHEIGHT, see
Table 2 for explanations of acronyms) and timing (PEAKDAY) are defined as the maximum chla value





at the corresponding day-of-year, respectively. Two threshold-based metrics and one distribution-fit-based metric were calculated:

1) Chla concentration exceeding a fixed-threshold of $5 \, \mathrm{mg \, m^{-3}}$ was defined as bloom by Fleming and Kaitala (2006), further referred to as `const5`. A 21-day centred-running-mean filter was used to keep results comparable to other metrics (Fleming and Kaitala (2006): 7-day centred-running-median filter).

2) Siegel et al. (2002) proposed a variable-threshold metric based on the 5 %-above-median concentration, but reported small quantitative differences for thresholds between 1 and 30 %-above-median. Their threshold is based on the complete annual cycle, while here only the spring bloom period from day-of-year 31 to 160 is considered. We refer to this metric as `median5`.

3) Distributions proposed to describe bloom phenology include shifted-Gaussian (Platt et al., 2009), Gamma (Vargas et al., 2009), and Weibull distributions (Rolinski et al., 2007). While the shifted Gaussian is symmetric in shape, Gamma distributions allow for different slopes of bloom rise and decline. In addition, Weibull functions recognize non-zero offsets before and after the bloom phase. The latter has proven essential to obtain a good fit for the transition phase between spring and summer bloom. A modified Weibull-function, as proposed by Rolinski et al. (2007), was fitted non-linearly to the preprocessed and scaled (to a range of zero to one) chla concentrations. The bloom initiation and end are defined as the $10^{th}$ and $90^{th}$ percentiles before and after the bloom peak, respectively. This metric is further referred to as `weibull`.

For each metric, bloom initiation, peak, and end dates (STARTDAY, PEAKDAY, and ENDDAY) were extracted from the data set. Based on these dates, bloom duration (DURATION), concentration average (CONCAVG), and the sum of daily chla concentrations (BLOOMIDX) were calculated. The latter was proposed by Fleming and Kaitala (2006) to characterize bloom intensity.

### 2.5 Principal Component Analysis

Principal component analysis (PCA) was carried out to attribute seasonally detrended meteorological conditions (SST, PAR, WIND) and nutrient concentrations (NUT-PEAKVALUE, NUT-MINVALUE) to the inter-annual variability in bloom intensity (BLOOMIDX, CONCAVG, PEAKHEIGHT) and timing (STARTDAY and PEAKDAY, DURATION). Outliers were defined for each parameter as departure by more than 3 standard deviations from the parameter mean, and replaced with the region-median. Z-score normalization (subtraction of mean, division by standard deviation) was carried out on a per-region basis. For 30 out of 225 combinations of sea region, year, and bloom metric, ferrybox records started after blooms had initiated. Such cases were identified 9 times for bloom metric `const5`, 15 times for `median5`, and 5 times for `weibull`. Corresponding bloom start days were replaced by the median value for the region over the 15 years studied during subsequent calculation of bloom phenology trends. Region-equalized, zero-mean and unit-variance data were then subjected to the PCA function in the python framework scikit-learn (Pedregosa and Varoquaux, 2011).



## 3 Results

### 3.1 Quality-controlled Chlorophyll-*a* Concentration Time Series

The Alg@line ferrybox systems collected over $9.5 \times 10^6$ observations between 2000 and 2014, of which $3.8 \times 10^6$ observations were sampled during spring (day-of-year 31 to 160). Availability and rejection rates for each quality control parameter are listed in Table 1. In total, quality control procedures removed 4.55 % of all observations.

Determination of the fluorescence yield was supported by an 'adaptive regression' method. Where necessary ($R^2 < 0.3$ or $p > 1$), it selected the subset of bottle-sampled and laboratory-analysed chla concentrations that yielded the best linear fit to chla fluorescence observations. This procedure allowed to successfully fit 318 (98 %) of the total 324 transects with bottle samples. Only 266 (82 %) transects could have been used without applying this technique.

Figure 3A shows normalized fluorescence observations as a function of sampling time-of-day. Results are presented separately for summer (May to August), winter (November to February) and the transition periods (autumn, spring). Diurnal variability was most pronounced in summer, when the fluorescence signal varied on average 50 % over the course of a day. In winter and during the transition periods (spring, autumn) the effect was less pronounced, although a diurnal variability of 35 and 38 % is still contained in the respective fluorescence signals. This seasonal effect is likely caused by variations in average irradiance intensity, which are modulated primarily by sun elevation, but also by atmospheric conditions (e.g. cloud cover, aerosol optical thickness) and optical properties of the water body (e.g. ice cover, attenuation). Figure 3B depicts normalized fluorescence as a function of solar elevation. In this representation seasonal differences in diurnal variability are essentially absent and the correspondence between solar elevation and average fluorescence response was approximately linear for daytime observations.

### 3.2 Bloom Intensity and Timing

Blooms generally developed first in the south and progressed towards the north (see Fig. 4 and Table 3). Bloom peak timing (not influenced by choice of metric) followed this pattern, as did metric-dependent bloom start and end dates. The fixed-threshold bloom metric `const5` suggested longer blooms in high-biomass sea areas like the GOF, compared to low-biomass areas such as the SBS. The variable-threshold metric `median5` applies area-specific bloom thresholds (NBP: $3.52\,\mathrm{mg\,m^{-3}}$, GOF: $4.95\,\mathrm{mg\,m^{-3}}$, GOT: $2.51\,\mathrm{mg\,m^{-3}}$, SBS: $2.62\,\mathrm{mg\,m^{-3}}$, BOM: $4.02\,\mathrm{mg\,m^{-3}}$) and resulted in approximately stable bloom durations for all sea areas. The `weibull` metric, which is not sensitive to absolute bloom intensity, also resulted in comparable bloom durations for all sea areas. The year-to-year variability of start, peak, and end days generally increased towards the south for all metrics.

Spring bloom intensity was described by three parameters: the metric-independent bloom peak concentration (PEAKHEIGHT), the chla concentration average during bloom conditions (CONCAVG),



and the sum of daily chla concentrations over the bloom period (BLOOMIDX). Similar patterns were
observed for all these parameters and bloom metrics, as illustrated in Fig. 5. The highest bloom
intensity was found in the GOF and NBP, followed by the BOM. Low-intensity blooms were ob-
served in the SBP and the GOT. Variability was generally proportional to bloom intensity, highest
in the high-biomass and coastal GOF and BOM. Variability in BLOOMIDX was comparable to that
in PEAKHEIGHT, while CONCAVG was considerably more stable. All calculated bloom phenology
parameters can be found in the supplementary material.

### 3.3 Trends

Figure 6 shows mean-normalized (subtraction of area-average concentration) CONCAVG and
PEAKHEIGHT for all sea areas combined as a function of bloom year. PEAKHEIGHT is inde-
pendent of bloom metric and shows a highly significant ($R^2 = 0.12, p \ll 0.01$) negative trend of
$-0.30 \pm 0.10\,\mathrm{mg\,m^{-3}\,yr^{-1}}$. CONCAVG is dependent on bloom start and end days and was therefore
calculated for all applied metrics. Statistically significant, negative trends resulted from all metrics:
$-0.12 \pm 0.04\,\mathrm{mg\,m^{-3}\,yr^{-1}}$ for `const5` ($R^2 = 0.11, p \ll 0.01$), $-0.11 \pm 0.05\,\mathrm{mg\,m^{-3}\,yr^{-1}}$ for
`median5` ($R^2 = 0.12, p < 0.05$), and $-0.22 \pm 0.07\,\mathrm{mg\,m^{-3}\,yr^{-1}}$ for `weibull` ($R^2 = 0.11, p \ll$
0.01).

No significant trends were found for BLOOMIDX, STARTDAY, and PEAKDAY with any of
the applied metrics, while ENDDAY showed weakly correlated but statistically significant ($R^2 =
0.06, 0.08, p < 0.05$) positive trends for `const5` and `weibull` with slopes $0.6$, $0.7 \pm 0.3\,\mathrm{d\,yr^{-1}}$,
respectively.

Bloom duration resulting from the `weibull` metric stands out in the result set with a positive
trend of $1.04 \pm 0.20\,\mathrm{d\,yr^{-1}}$ ($R^2 = 0.28, p \ll 0.01$, Fig. 7). No significant trend in bloom duration
was found for any fixed- or variable-threshold metric.

Peak nutrient concentrations showed no significant trend, in contrast to post-bloom nutrient con-
centrations with a highly significant, negative trend $-0.020 \pm 0.004\,\mathrm{\mu mol\,L^{-1}\,yr^{-1}}$ ($R^2 = 0.23, p \ll$
0.01). Peak nutrient concentration timing shifted to earlier dates ($-0.7 \pm 0.3\,\mathrm{d\,yr^{-1}}$ ($R^2 = 0.06, p <$
0.05)), while the 25 %-of-peak-value was reached progressively later ($0.67 \pm 0.31\,\mathrm{d\,yr^{-1}}$, ($R^2 =
0.06, p < 0.05$)). No significant trends were found for nutrient depletion slope, 50 %-of-peak-value-
timing, or day of minimal nutrient concentrations.

### 3.4 Inter-annual Variability

Pre-bloom nutrient concentrations were positively correlated to bloom peak height (no normal-
ization, $R^2 = 0.39, p \ll 0.01$) and concentration average (no normalization, $R^2 = 0.37 - 0.57, p \ll$
0.01, depending on metric). Surprisingly, after applying area-wise mean and variance (z-score) nor-
malization, a negative correlation was found for PEAKHEIGHT ($R^2 = 0.11, p \ll 0.01$, metric inde-
pendent) and CONCAVG ($R^2 = 0.12, 0.11, p \ll 0.01$ for `const5` and `weibull`, respectively).



Nutrient-depletion timing, specifically NUT-DEPLDAY-50, was positively correlated to the bloom peak day ($R^2 = 0.47, p \ll 0.01$), as well as bloom-averaged, detrended PAR-levels ($R^2 = 0.14 - 0.29, p \ll 0.01$). Average wind speed and PAR were negatively correlated during bloom conditions ($R^2 = 0.10 - 0.23, p \ll 0.01$). The bloom timing parameters (STARTDAY, PEAKDAY, ENDDAY) were weakly but statistically significantly inter-correlated (results not shown).

PCA scores and loadings of the first three principal components (PC) are shown as biplots in Fig. 8. The first PC is dominated by negative correlations to bloom intensity parameters (PEAKHEIGHT, CONCAVG, BLOOMIDX). This component is positively correlated to pre-bloom nutrient concentration (NUT-PEAKVALUE) and bloom duration, illustrating that bloom intensity is affected by pre-bloom nutrient availability. The second PC is linked to bloom timing, with strong positive correlations to STARTDAY and PEAKDAY. Correlations to PAR (positive), SST (positive), and WIND (negative) suggest that weather conditions affect bloom timing. Bloom duration is positively correlated to the third PC, as well as to BLOOMIDX. Additional negative correlations to NUT-MINVALUE and WIND, as well as a positive correlation to PAR, suggest a link between favourable meteorological conditions (low wind-mixing, high light level) and efficient nutrient depletion.

## 4  Discussion

Trends in spring bloom phenology can be interpreted as responses to nutrient reduction as well as to slowly acting environmental processes, such as climate change. To disentangle or even quantify these trends, suitable observation platforms and subsequent analytical approaches must be chosen. We present evidence that fundamental challenges of ferrybox observations can be overcome to yield an internally consistent data source. Subsequently, the behaviour of commonly used bloom metrics in presence of decadal trends can be scrutinized in the context of previously reported system knowledge. Finally, we attempt to disentangle the effects of nutrient availability and meteorological conditions on inter-annual variability in bloom phenology.

### 4.1  Automated Processing of Ferrybox Observations

Thresholds for speed, flow rate, and data variability were iteratively adjusted to the data set and might therefore not apply directly to other ferrybox implementations. Particularly flow rate derived from differences in line and hull temperature likely requires tuning to each ferrybox installation. Transect-wise normalization of the quality controlled fluorescence data was adequate to consistently interpret observations collected by different generations of instrumentation. However, this approach crucially depends on continuous temporal coverage of reference measurements for calibration to chla concentrations. Adaptive regression analysis improved the handling of statistical outliers which would otherwise hamper determination of fluorescence yield, while transects for which no bottle samples are available were corrected with an interpolated fluorescence yield derived from the closest bottle-





sampled transects. The present procedure allows for automated and reproducible processing which is an improvement over manual quality control. Applying the proposed interpolated fluorescence yield helps in reprocessing and long-term data analysis of ferrybox fluorescence observations to better represent natural variability.

### 4.2 Variability in Fluorescence Yield

Fluorescence diurnal variability showed low seasonal dependence after accounting for solar elevation. Unsurprisingly, light intensity is the predominant factor in Baltic Sea phytoplankton fluorescence yield variability. Other seasonal differences in fluorescence response can be attributed to typically higher cloud cover in winter compared to summer and spring/autumn, which was not accounted for in our analysis. The seasonal cycle of species composition, from dinoflagelate and diatom dominated spring communities (Klais et al., 2011) to cyanobacterial summer bloom (Kahru and Elmgren, 2014), influenced fluorescence yield considerably less than diel cycles.

The diurnal variability in fluorescence response of 50 % during an average summer day is within a range of earlier findings, e.g. 66 % ($\pm$33%) for near surface observations in upwelled waters of the equatorial Pacific reported by Dandonneau and Neveux (1997) or 30 % for near-surface seaglider observations in Northeast Pacific waters off the Washington coast, USA (Sackmann et al., 2008), although differences in normalization impede direct comparison. The sampling depth of 5 m for Alg@line systems and the high attenuation of the Baltic Sea in comparison to clear Pacific Ocean waters cause lower diurnal variability.

In this study fluorescence observations during spring, when diurnal variability reached on average 38 %, were binned for five large Baltic Sea areas. At a typical cruising speed of approximately 23 kn each sea area is sampled for at least several hours. This limits the influence of diurnal variability in fluorescence yield along a transect, which is therefore of lesser relevance for the present study. However, if fluorescence measurements were to be quantitatively evaluated at a higher spatial resolution, variable fluorescence yield should be accounted for. Analysis of signal-coherence (Groetsch et al., 2014) offers an alternative to quantitative interpretation of fluorescence observations and can be used to qualitatively detect cyanobacterial surface bloom. If light history is known, e.g. from a dedicated irradiance sensor, a correction of diurnal fluorescence yield variability might be possible and further research in this direction is recommended.

### 4.3 Spring Bloom Timing and Intensity

The presented bloom phenology expands the time series presented by Fleming and Kaitala (2006) and is in good agreement for the overlapping period (2000 - 2004) when comparing the `const5` metric results. Remaining differences are likely due to quality-control and pre-processing procedures on the fluorescence records. In their work, the authors reported for GOF, NBP, and the Arkona Sea that bloom typically started in the south and ended in the north, while bloom intensity increased



towards the north. These observations are confirmed here. Sea areas not covered in Fleming and Kaitala (2006), e.g the high-biomass BOM and low-biomass SBP and GOT, followed the reported
south-north trend in bloom development. Present results also support and expand the findings of Fennel (1999), who showed with simulations and monitoring data from 1994-1996 for the Western Baltic Sea that surface heating in early spring needs to overcome the temperature of maximum density to repress convective mixing and allow spring bloom to emerge. The temperature of maximum density increases with decreasing salinity, so that convective mixing is sustained longer in less saline
northern Baltic Sea waters when spring temperature is on the rise. At the same time, incident solar radiation increases slower in the north due to lower solar elevation.

#### 4.4 Trends

Interannual variability in coastal systems exceeds long-term trends by orders of magnitude (Cole et al., 2012). Consequently, trends were observed at relatively low coefficients of correlation. The
importance of appropriate data pre-processing and gap-handling (e.g Cole et al., 2012; Racault et al., 2014) and choice of metric (Ferreira et al., 2014) has been emphasized in literature and is further demonstrated by the present analysis. Robustness of the reported decadal trends is documented by high statistical significance levels ($p \ll 0.01$, Figs. 7 and 6), which were supported by spatially binning phenology parameters from all examined Baltic Sea areas. Similar trends were observed earlier
for individual Baltic Sea areas, however, usually outside 95 % confidence intervals (e.g Wasmund and Uhlig, 2003).

    Łysiak Pastuszak et al. (2014) reported stable or increasing chla concentrations for the period 2007-2011 in several Baltic Sea areas despite signs of declining nutrient concentrations. More recently, eutrophication trend reversal and oligotrophication processes were reported by Andersen et al.
(2015), based on analysis of 112 years of consolidated Baltic Sea observations. Both reports considered surface-layer chla concentration in summer as one of the direct indicators for eutrophication, but did not include spring bloom in their assessment. The time series for 2000-2014 that we present here fills this gap: a negative trend in bloom intensity was found also for spring bloom, providing further evidence for their hypothesis.

The concentration distribution-ratios on which the Weibull-metric is based are calculated for each bloom individually, in contrast to the thresholds of `const5` and `median5` that are fixed for the complete time series (see Fig. 1). Threshold-based metrics revealed no significant trends in bloom duration, while Weibull-distribution metrics showed a highly significant, positive trend. These two contrasting results nevertheless support the conclusion that spring blooms in the Baltic Sea have
become longer, while chla peak and average concentration levels declined. This 'flattening' of the concentration distribution is supported by the absence of a trend in time-integrated biomass BLOO-MIDX and by shifts in nutrient concentration timing (earlier nutrient peak concentration, later 25 %-of-peak-value day). These results indicate that annually generated spring bloom biomass has not





changed significantly over the study period, in contrast to bloom timing. Kahru and Elmgren (2014)
found a similar development for cyanobacterial summer surface bloom, and reported decadal oscillations, yet no long-term trend, of surface area covered by cyanobacteria in the period 1979-2013. In the same period, summer bloom initiation moved to earlier dates by $-0.6\,\mathrm{d\,yr^{-1}}$. These results suggest that the gap has decreased between dinoflagelate- and diatom-dominated spring bloom and cyanobacterial summer bloom.

Our findings emphasize that bloom timing is an essential indicator to monitor marine ecosystem dynamics, and thus eutrophication status. Crucial for deriving bloom timing trends are observations at high temporal resolution and choice of bloom metrics. Eutrophication status assessment frameworks such as HEAT3.0 (Andersen et al., 2015) may be adapted to embrace available high-frequency data sources to include bloom timing in their analysis. Ecosystem models of the Baltic Sea and other coastal or inland systems can also use the presented results for validation and to enhance their predictive capabilities.

### 4.5 Environmental Forcing

Gradually decreasing nutrient concentrations (Łysiak Pastuszak et al., 2014; Andersen et al., 2015), as well as rising average air- and sea-surface temperatures (Omstedt et al., 2004; Borsenkova et al., 2013) have been reported for recent years, corresponding to a combination of nutrient-reduction efforts and global climate change. Several scenarios for future change are plausible (Duarte et al., 2009) but extrapolation of present results is beyond the scope of this study. However, we attempt to attribute the observed bloom phenology shifts to reported changes in environmental drivers.

Winter-time nutrient concentration and bloom intensity were positively correlated if no spatial normalization was applied. This supports the paradigm that the first-order driver of bloom intensity is nutrient availability. Therefore, lacking other explanations, we attribute the reported negative trend in bloom peak concentration to declining nutrient concentrations. First-order spatial trends in bloom intensity and timing can be removed by an area-wise z-score normalization, which effectively constrains the analysis to inter-annual variability. After this normalization both regression and PCA resulted in negative correlation between winter-time nutrient concentration and bloom intensity. This negative feedback can be understood as a subtle interaction between meteorological forcing and nutrient supply: strong wind-forced mixing can cause upwelling of deep, nutrient rich waters to surface layers. Wind speed, however, was found to be negatively correlated to the prevalent light level, as well as to bloom duration and bloom index. Therefore, in years when additional nutrients are available due to strong wind forced mixing, low-light regimes that can hamper bloom development also prevail.

Bloom duration co-varied primarily with weather conditions, e.g. high irradiance levels and low wind speeds were frequently observed for long-lasting blooms (and vice versa). Although the same pattern was observed for bloom timing, no trend was found for bloom start- and peak-day. Increas-



ingly favourable meteorological conditions in late bloom phases are thus a likely driver for the observed increase in bloom duration. Similar weather-driven modulations of bloom timing were reported earlier (Fleming and Kaitala, 2006; Meier et al., 2011; Neumann et al., 2012) for spring, and especially cyanobacterial summer bloom (Wasmund, 1997; Kanoshina et al., 2003; Wynne et al., 2010, 2011).

## 5  Conclusions

A Baltic Sea spring bloom phenology was derived from 15 years of automated ferrybox chla fluorescence observations. Procedures for automated quality-control and processing were introduced and uncertainty due to diurnal variability in phytoplankton fluorescence response was quantified. Both promote increased use of ferrybox observations for scientific research and monitoring purposes, such as the periodic HELCOM eutrophication status assessments. Negative trends in spring bloom peak- and average-concentration were found and an increase in bloom duration was derived from conceptually differing bloom metrics. Inter-annual variability in bloom intensity was primarily linked to nutrient availability, while bloom timing and duration was found to be related to meteorological conditions. In the future, these findings might allow to better disentangle ecosystem response to changing nutrient availability and climatic conditions.

*Acknowledgements.* The authors thank the Alg@line consortium, specifically scientists and technical personnel at SYKE (and formerly FIMR), for the ferrybox in situ data set. Acknowledgement is made to ECMWF for the use of their ERA-Interim data set in this research. MAE, SWMP and PMMG were co-funded by the European Community Seventh Framework Programme under grant agreement 607325 AQUA-USERS, and grant agreement 313256 GLaSS. PMMG also received support from EC/IAPP project WaterS (Grant 251527).



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



**Table 1.** Quality control flag definitions and statistics. Observations were excluded if any of the flags exceeded the respective threshold. Absolute temperature difference is measured between the water intake and the flow-through sensors. Availability and reject rates were calculated relative to the total number of data points.

|  | Sign | Threshold | Availability [%] | Rejection Rate [%] |
|---|---|---|---|---|
| Speed, [kn] | < | 5 | 100 | 1.33 |
| Flow, [L min$^{-1}$] | < | 0.3 | 35.95 | 1.38 |
| Abs. Temp. Diff. [°C] | > | 2 | 67.17 | 2.12 |
| STD Latitude, [°] | <, > | $1e^{-4}$, 0.5 | 100 | 0.96 |
| STD CHLa Fl., [mg m$^{-3}$] | < | $1e^{-4}$ | 87.65 | 0.75 |
| All |  |  |  | 4.55 |

**Table 2.** Description and acronyms of bloom phenology, nutrient, and meteorological parameters that were used in the trend and multi-variate analysis.

| Parameter | Unit | Description |
|---|---|---|
| BLOOMIDX | mg d m$^{-3}$ | Integrated chlorophyll-$a$ concentration during bloom |
| CONCAVG | mg m$^{-3}$ | Average (mean) chlorophyll-$a$ concentration during bloom |
| PEAKHEIGHT | mg m$^{-3}$ | Highest chlorophyll-$a$ concentration during bloom |
| STARTDAY | Julian Day | Bloom start day |
| PEAKDAY | Julian Day | Bloom peak day |
| ENDDAY | Julian Day | Bloom end day |
| NUT-MINVALUE | µmol L$^{-1}$ | Nutrient concentration at end of bloom |
| NUT-MINDAY-50 | Julian Day | Day when nutrients equalled 50 % of NUT-PEAKVALUE |
| NUT-PEAKVALUE | µmol L$^{-1}$ | Pre-bloom (winter-time) nutrient concentration |
| NUT-PEAKDAY | Julian Day | Day of NUT-PEAKVALUE |
| NUT-DEPLAY-25 | Julian Day | Day when nutrient concentration equalled 25 % of NUT-PEAKVALUE |
| NUT-DEPLAY-50 | Julian Day | Day when nutrients concentration equalled 50 % of NUT-PEAKVALUE |
| NUT-SLOPE | µmol L$^{-1}$ d$^{-1}$ | Rate of nutrient depletion between 75 and 25 % of NUT-PEAKVALUE |
| PAR | W m$^{-2}$ d$^{-1}$ | Average (seasonally detrended) photosynthetically active radiation level |
| SST | °C | Average (seasonally detrended) sea surface temperature |
| WIND | m s$^{-1}$ | Average (seasonally detrended) wind speed |




**Table 3.** Bloom timing and intensity for each investigated sea area (Figure 2) and for all applied bloom metrics.

| Parameter | Sea Area / Metric | BOM Mean | BOM Std | SBP Mean | SBP Std | GOT Mean | GOT Std | NBP Mean | NBP Std | GOF Mean | GOF Std |
|---|---|---|---|---|---|---|---|---|---|---|---|
| STARTDAY, [Julian Day] | const5 | 68 | 9.3 | 87 | 13.6 | 95 | 8 | 89 | 5.6 | 81 | 8.4 |
| | median5 | 65 | 9.4 | 73 | 7.3 | 83 | 9.4 | 86 | 4.4 | 81 | 8.5 |
| | weibull | 64 | 12.7 | 73 | 7.2 | 84 | 6 | 87 | 4.1 | 89 | 4.7 |
| PEAKDAY, [Julian Day] | all metrics | 75 | 14.7 | 92 | 14.9 | 106 | 7.4 | 108 | 4.4 | 112 | 4.7 |
| ENDDAY, [Julian Day] | const5 | 95 | 15.2 | 102 | 12.5 | 118 | 10.7 | 130 | 5.2 | 143 | 5.6 |
| | median5 | 107 | 20.3 | 115 | 13.1 | 133 | 7.1 | 142 | 3.3 | 143 | 5.2 |
| | weibull | 94 | 18.4 | 116 | 15.4 | 128 | 9 | 126 | 5.6 | 132 | 5.7 |
| DURATION, [d] | const5 | 35 | 12.5 | 16 | 12.4 | 23 | 13.6 | 41 | 6.4 | 62 | 11.7 |
| | median5 | 54 | 18.9 | 46 | 15.4 | 47 | 8.8 | 54 | 4.4 | 62 | 12.3 |
| | weibull | 36 | 10.8 | 43 | 16.4 | 44 | 10.9 | 40 | 6.1 | 43 | 6.7 |
| BLOOMIDX, [mg dm$^{-3}$] | const5 | 283 | 167.6 | 98.4 | 77.1 | 162.4 | 114.2 | 352.4 | 84.9 | 691.6 | 157.6 |
| | median5 | 334.3 | 135.1 | 197 | 92.6 | 224.5 | 77.2 | 386.9 | 72.9 | 694 | 165.9 |
| | weibull | 356 | 178.5 | 196.7 | 74.1 | 232.9 | 64.1 | 340.1 | 62.1 | 673.7 | 175.9 |
| CONCAVG, [mg m$^{-3}$] | const5 | 7.3 | 2.1 | 5.3 | 0.9 | 6.2 | 1.3 | 8.4 | 1.6 | 11.7 | 2.2 |
| | median5 | 6 | 1.2 | 4.1 | 0.7 | 4.6 | 1.1 | 7 | 1.1 | 11.7 | 2.3 |
| | weibull | 9.9 | 4.3 | 4.6 | 1.1 | 5.5 | 1.7 | 8.5 | 2.2 | 13.6 | 3.3 |
| PEAKHEIGHT, [mg m$^{-3}$] | all metrics | 12.3 | 5.2 | 6.1 | 1.7 | 7.2 | 2.3 | 11.3 | 2.9 | 20.2 | 5.7 |




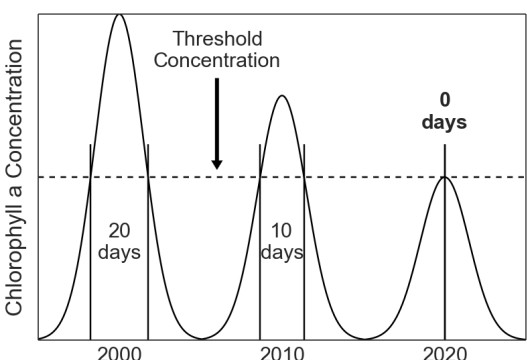

**Figure 1.** Illustration of threshold-based bloom metric behaviour when applied to a dataset with a negative peak concentration trend.

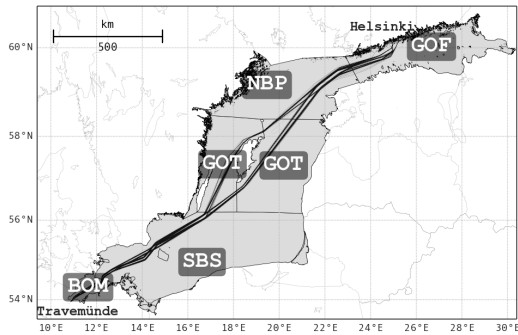

**Figure 2.** Transect of M/S *Finnmaid* and M/S *Finnpartner* through the Baltic Sea from Helsinki (Finland) to Travemünde (Germany) (v.v.). The following sea areas are considered in this study: the Western Gulf of Finland (GOF: >59.5 °N latitude, along transect), the Northern Baltic Proper (NBP: 58.4-59.5 °N latitude, along transect), the Western and Eastern Gotland basins (GOT: 56.2-58.4 °N latitude, along transect), the Southern Baltic Proper (SBP: 54.5-56.2 °N latitude, along transect) and the Bay of Mecklenburg (BOM: <54.5 °N latitude, along transect). Depending on weather conditions the North- or South of Gotland routes were sailed.




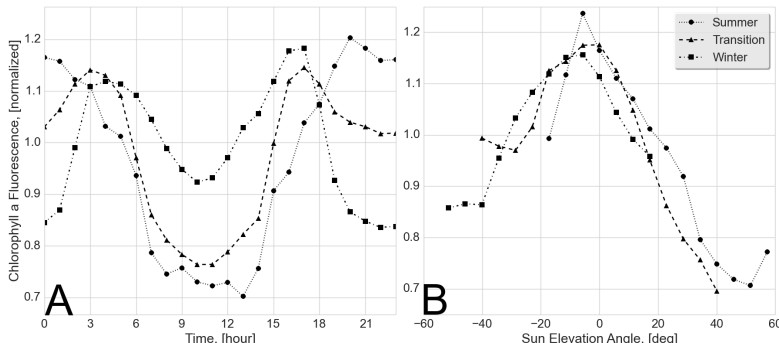

**Figure 3.** Diurnal variability in the chlorophyll-*a* fluorescence yield: (A) normalized (division by transect-mean) chlorophyll-*a* fluorescence observations plotted against time-of-day, and (B) sun elevation angle. The analysis was carried out on four subsets: winter (November - February), summer (May - August), and transition periods (March, April, September, October) using all ferrybox observations along the routes shown in Figure 2.

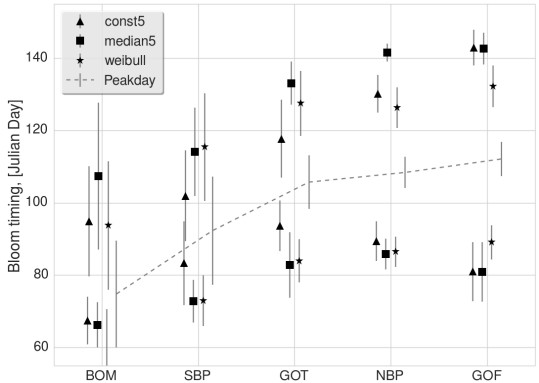

**Figure 4.** Bloom timing (bloom start, peak, and end day) for each sea area along the routes in Figure 2, averaged over the period 2000 to 2014, and for all applied bloom metrics. Whiskers indicate standard deviations over the 15-year study period. The bloom peak-day is independent of the chosen metric, and thus plotted separately. The sea areas are ordered by latitude, from south to north.





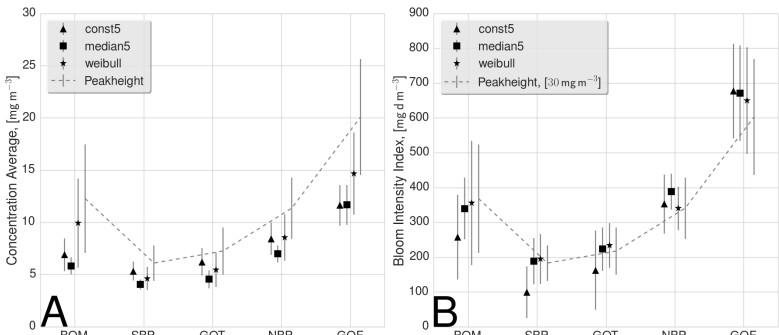

**Figure 5.** (A) Concentration average and (B) bloom intensity index for each sea area along the routes in Figure 2, averaged over the years 2000 to 2014, and for all applied bloom metrics. Whiskers indicate standard deviations over the 15-year study period. The sea areas are ordered by latitude. The metric-independent bloom peak concentration is added in both plots for visual comparison.

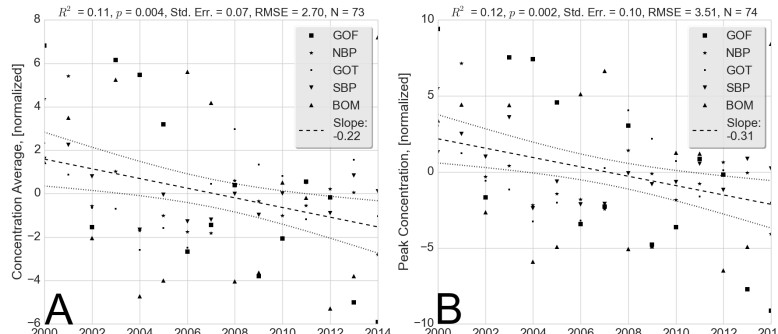

**Figure 6.** (A) Decadal trend of average (CONCAVG) and (B) peak (PEAKHEIGHT) chlorophyll-*a* concentration during bloom conditions, derived from the Weibull-distribtion metric. Concentrations were normalized prior to regression (subtraction of area-average concentration). Dashed lines indicate the trend line (bold) and its confidence intervals (5 %, small dashes).





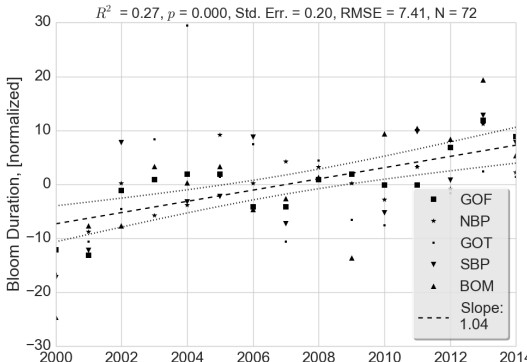

**Figure 7.** Decadal trend of bloom duration (DURATION), calculated with the Weibull-distribution metric. Durations were normalized prior to regression (subtraction of area-average duration). Dashed lines indicate the trend line (bold) and its confidence intervals (5 %, small dashes).

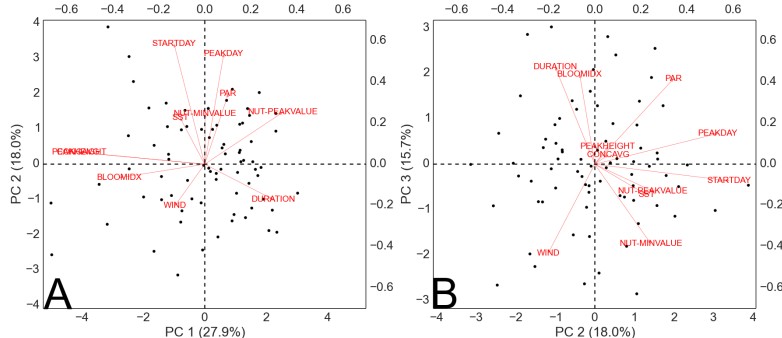

**Figure 8.** Principal component analysis bi-plots: arrows indicate correlation of a parameter with the principal components (bottom- and left-axes, percentages refer to the variability explained by the principal component), and black dots indicate scores of individual observations (top- and right-axes) on the principal components (A: component 1 and 2, B: component 2 and 3).