# Peer review of "Spring Blooms in the Baltic Sea have weakened but lengthened from 2000 to 2014"

_Biogeosciences, 2015_

## Referee Comment (RC1) · Anonymous Referee #1 · 5 Feb 2016

The manuscript aims at the analysis of chlorophyll data from the Baltic Sea. The authors use primarily data collected by automated systems onboard of ships of opportunity. Finally they end up with some trends detected in spring bloom timing and duration for the last 15 years.

In my view, an important part of the manuscript is the described method how to handle, validate and analyze the Ferrybox data. It is inspiring for further analysis and using the tremendous pool of information of the marine environment.

In the discussion section, the results of spring bloom dynamics are attributed to possible drivers: Nutrient availability and meteorological forcing. Consistent with the approach, the attribution is based on statistical methods. This could be assisted by advanced ecosystem models in future research. However, this is beyond this study.
[Figure]

Altogether, in my opinion this is a well done study and should be published after minor revision.

Remarks to authors:

Could you please provide some information on the source(s) of nutrient data in chap. 2.3.

The title is not much appealing to me. However, I don't have any good suggestion. Probably you may include alg@line.

---

## Referee Comment (RC2) · Anonymous Referee #2 · 15 Feb 2016

This manuscript presents an analysis of 15-year time series of chlorophyll fluorescence at six spatially averaged regions of the Baltic Sea. Fluorescence observations come from ships of opportunity and are used to evaluate trends on phytoplankton phenology. The detailed description of the systematic quality control methods for this type of observational datasets is an important contribution that ensures replicability of the analysis. The authors discuss trends between 2000 and 2014 using different phenological metrics. The manuscript is well written and interesting patterns in the region are brought to attention. I would recommend the following revisions to improve the clarity of methods and discussion:

1. Lines 90-95 / Figure 1. Text mentions that "any threshold-based metric" would introduce artificial trends in bloom duration. This is a clear problem for "fixed threshold" metrics, but not for "variable thresholds" as Siegel et al. (2002), which is later intro-

duced. Furthermore, results using the fixed threshold const5 show a negative trend in peak concentrations, but no significant trend in bloom duration. This seems a somewhat inconsistent. Further discussion would help clarify why the expected artificial trends do not occur.

2. It is not clear to me whether median5 (Siegel et al. 2002) is calculated for each individual annual median or for all years together. The latter would indeed produce a fixed threshold for each region (see previous comment). That detail is unclear in Siegel et al. 2002 as well, but see Henson et al. 2009 (Decadal variability in North Atlantic phytoplankton blooms – J. Geophys. Res.) and Brody et al. 2013 (A comparison of methods to determine phytoplankton bloom initiation – J. Geophys. Res.).

3. Lines 195-200: Day-of-year 31 is January 1?

4. Why was the time frame between day 31 -160 selected? Is it possible that nutrient peak concentration occur prior to the minimum date considered? A shift to earlier peak nutrient concentrations is mentioned, but results of the nutrient metrics are not presented. I suggest extending Table 3 and/or including plots to support this.

5. Lines 230-235: In 30 out 225 data combinations there were no ferrybox observations to properly identify bloom initiation. In these cases, bloom initiation date was replaced by the median value. It is not clear if this treatment was used only for the principal component analysis or the regressions as well. Cases identified by each timing method only account for 29 (const5:9, median5: 15, weibull: 5). I find it also unclear how these methods identified that the bloom had already started. A few words to clarify would be useful.

6. The time series analyzed is relatively short to claim long-term trends, especially when considering the large interannual variability observed in all of the metrics. A study between 1979-2013 where decadal-oscillations were found is mentioned in the text. I would recommend extending the discussion a bit to include how that analysis compares with this one during the same time frame.

7. The final discussion and conclusions attribute the declining trend in bloom peak concentration to declining nutrient concentrations; however, no decline in winter nutrient concentrations (as estimated here) is reported. The conclusion is based on literature considerations and the "lack(ing) of other explanations". I think this pattern is quite interesting and an alternative explanation may be supported by the results here presented. The authors report a shift in peak nutrient concentration to earlier dates and a strong correlation between winter nutrient concentration and bloom peak magnitude. Earlier increases in nutrient concentrations mean that nutrient limitation is alleviated earlier during the year, when light limitation might still be strong. As the year progresses and light limitation is alleviated, a fraction of the nutrients has been already consumed. The nutrient concentration "available for blooming" would then not be equal to the winter maximum, but lower than it. That would produce a decrease in the bloom peak magnitude, an apparent extend in bloom duration, but no change in total chlorophyll during the bloom (also reported). This is just a quick idea and might be better captured by rate-of-change metrics of bloom phenology, which are mentioned in the introduction, but not used in the analysis. As I mentioned before, I think it is important to include the nutrient concentrations results in the manuscript to better support its conclusions. I would also suggest including the actual time series (environmental factors and fluorescence) as part of supplementary material.

---

## Author Comment (AC1) · 18 Apr 2016

1. Could you please provide some information on the source(s) of nutrient data in chap. 2.3.

Authors' response: These concentrations were derived from laboratory analysis of bottle samples that were regularly collected along the transect (for further detail see section 2.1. in the manuscript). We propose to add this information to section 2.3 to clarify this point.

2. The title is not much appealing to me. However, I don't have any good suggestion. Probably you may include alg@line.

Authors' response: The title was chosen to highlight the scope of the paper (phenological study of Baltic Sea spring bloom) rather than the methods used to obtain this value-adding dataset. We hope that this will draw readers to the paper who would not normally consider ship-of-opportunity pigment fluorescence data for this type of analysis. We suggest to include an early reference to the 'Alg@line' network in the abstract, e.g. 'Phytoplankton spring bloom phenology was derived from a 15-year time-series (2000-2014) of ship-of-opportunity chlorophyll-a fluorescence observations (collected in the Alg@line network) in the Baltic Sea. '

---

## Author Response (AR1)

**Response to reviewers on 'Spring Blooms in the Baltic Sea have weakened but lengthened from 2000 to 2014' by P. M. M. Groetsch et al.**

We would like to thank the two reviewers for the comments provided. After implementing revision changes the manuscript was proofread leading to a number of minor corrections (punctuation, grammar, clarity). An error in section 2.1 was corrected were two numbers were switched around (percentages of routes sailed east and west of Gotland). We also introduced some minor simplifications to the figures (no grid lines, no legend shading, open symbols).

**Anonymous Referee #1**

1. Could you please provide some information on the source(s) of nutrient data in chap. 2.3.

*Authors' response: These concentrations were derived from laboratory analysis of bottle samples that were regularly collected along the transect (for further detail see section 2.1. in the manuscript). We added this information to section 2.3 to clarify this point.*

2. The title is not much appealing to me. However, I don't have any good suggestion. Probably you may include alg@line.

*Authors' response: The title was chosen to highlight the scope of the paper (phenological study of Baltic Sea spring bloom) rather than the methods used to obtain this value-adding dataset. We hope that this will draw readers to the paper who would not normally consider ship-of-opportunity pigment fluorescence data for this type of analysis. We included an early reference to the 'Alg@line' network in the abstract:*
*'Phytoplankton spring bloom phenology was derived from a 15-year time-series (2000-2014) of ship-of-opportunity chlorophyll-a fluorescence observations collected in the Baltic Sea through the Alg@line network'*

**Anonymous Referee #2**

1. Lines 90-95 / Figure 1. Text mentions that "any threshold-based metric" would introduce artificial trends in bloom duration. This is a clear problem for "fixed threshold" metrics, but not for "variable thresholds" as Siegel et al. (2002), which is later introduced.

*Authors' response: Please see our detailed response with question 2).*

Furthermore, results using the fixed threshold const5 show a negative trend in peak concentrations, but no significant trend in bloom duration. This seems a somewhat inconsistent. Further discussion would

help clarify why the expected artificial trends do not occur.

*Authors' response: Peak concentrations are derived the same for all metrics, thus the negative trend is not metric dependent. Indeed, no artificial negative trend in bloom duration was observed for any threshold-based (fixed, or derived from climatology; see question 2)) metrics. We argue that this observation can be expected if blooms also became longer over the study period. An independent confirmation of this hypothesis is the strong positive trend in bloom duration in the Weibull-metric results. This is now stated more explicitly in section 4.4:*

*'Thresholds of const5 and median5 are fixed for the whole time-series. The observed negative trend in peak concentration introduces an artificial negative trend in bloom duration due to a shortening of the part of the curve seen above the bloom threshold, but this is solely by decreased amplitude of the curve (see Fig. 1). Contrary to this expected behavior, however, const5 and median5 revealed no significant trends in bloom duration. This indicates that the anticipated negative trend was countered by a positive trend. The Weibull-metric is based on concentration distribution-ratios that are calculated for each bloom individually. Therefore Weibull-metric results for bloom duration are not sensitive to long-term trends in peak concentration. Weibull-distribution metrics confirmed a highly significant, positive trend in bloom duration. These two sets of results mutually support the conclusion that spring blooms in the Baltic Sea have become longer, while chla peak and average concentration levels declined.'*

2. It is not clear to me whether median5 (Siegel et al. 2002) is calculated for each individual annual median or for all years together. The latter would indeed produce a fixed threshold for each region (see previous comment). That detail is unclear in Siegel et al. 2002 as well, but see Henson et al. 2009 (Decadal variability in North Atlantic phytoplankton blooms – J. Geophys. Res.) and Brody et al. 2013 (A comparison of methods to determine phytoplankton bloom initiation – J. Geophys. Res.).

*Authors' response: Indeed we assumed that Siegel et al. 2002 referred to the climatological median, rather than the annual median. Brody et al. 2013, however, state that both thresholds can be applied:*

*'The threshold bloom initiation method was introduced for marine phenology studies in Siegel et al. [2002]. This method finds the yearly or climatological median of a chlorophyll time series, then identifies the bloom start date as the first point at which chlorophyll levels rise a certain percentage above the median.'*

*To make the distinction clear we changed the respective paragraph in the introduction:*
*'Figure 1 illustrates how a gradual decline (negative trend) in bloom peak concentration causes any metric based on fixed thresholds (e.g. derived from climatology or expert-judgement) to introduce an artificial negative trend in bloom duration. In contrast, metrics based on growth-rate, distribution, or annually derived thresholds yield a single bloom duration for the given example, because bloom intensity does not influence these metrics.'*

3. Lines 195-200: Day-of-year 31 is January 1?

*Authors' response: This is in error and should read 'day-of-year between 31 (31 January) and 160 (9 June).*

4. Why was the time frame between day 31 -160 selected? Is it possible that nutrient peak concentration occur prior to the minimum date considered? A shift to earlier peak nutrient concentrations is mentioned, but results of the nutrient metrics are not presented. I suggest extending Table 3 and/or including plots to support this.

*Authors' response:*
*The ship-of-opportunity (Alg@line ) measurements typically commenced in late January, which is why we chose 31 January as the start of our analysis. The end date was chosen such that it covers all spring bloom events in all basins but not summer bloom. We added this information after the first paragraph of section 2.4.*
*The nutrient peak concentration is closely related to the day of bloom initiation, which is typically at least a month later (see table 3). Of more concern is that for several years, Alg@line data collection commenced after bloom initiation. These data were consequently omitted from statistical analysis (replaced with multi-year median), including the nutrient statistics. Further detail on this issue is also given in response to question 7. Table 3 shows multi-year averages of calculated parameters, so we can not expand to trends in nutrient timing or intensity from these. Since these results are nevertheless available, we added all nutrient metric results to the appendix to aid future research.*

5. Lines 230-235: In 30 out 225 data combinations there were no ferrybox observations to properly identify bloom initiation. In these cases, bloom initiation date was replaced by the median value. It is not clear if this treatment was used only for the principal component analysis or the regressions as well. Cases identified by each timing method only account for 29 (const5:9, median5: 15, weibull: 5). I find it also unclear how these methods identified that the bloom had already started. A few words to clarify would be useful.

*Authors' response: Median-filling of missing dates was applied prior to both PCA and regression analysis. We assumed bloom initiated prior to Alg@line data collection if the first data point already satisfied the bloom criterion for a given metric. This is now stated more clearly already in section 2.4 instead of in section 2.5. The number of missed bloom initiation events is incorrectly stated as 15 for the median5 metric and should be changed to 16.*

6. The time series analyzed is relatively short to claim long-term trends, especially when considering the large interannual variability observed in all of the metrics. A study between 1979-2013 where decadal-oscillations were found is mentioned in the text. I would recommend extending the discussion a bit to include how that analysis compares with this one during the same time frame.

*Authors' response: The authors of the mentioned study (Kahru2014) describe surface accumulations of cyanobacterial summer bloom. Links between spring bloom and cyanobacterial summer bloom are certainly worthwhile exploring. However, the complex interactions between light- and nutrient-limited spring bloom, and largely wind-modulated cyanobacterial surface bloom accumulations seem out of scope for the present paper (and may quite possibly be too complex). In section 4.4. we acknowledge the finding of Kahru2014 that summer bloom initiation moved to earlier dates, and thus that the period between dinoflagelate- and diatom-dominated spring bloom and cyanobacterial summer bloom decreased. The following sentence was added to section 4.4 to clarify that we can neither prove nor disprove a decadal oscillation signal based on our time series:*
*'However, due to the shorter period covered here as compared to the time series presented by Kahru2014, it cannot be ruled out that the derived trends are caused by decadal oscillation.'*

7. The final discussion and conclusions attribute the declining trend in bloom peak concentration to declining nutrient concentrations; however, no decline in winter nutrient concentrations (as estimated here) is reported. The conclusion is based on literature considerations and the "lack(ing) of other explanations". I think this pattern is quite interesting and an alternative explanation may be supported by the results here presented. The authors report a shift in peak nutrient concentration to earlier dates and a strong correlation between winter nutrient concentration and bloom peak magnitude. Earlier increases in nutrient concentrations mean that nutrient limitation is alleviated earlier during the year, when light limitation might still be strong. As the year progresses and light limitation is alleviated, a fraction of the nutrients has been already consumed. The nutrient concentration "available for blooming" would then not be equal to the winter maximum, but lower than it. That would produce a decrease in the bloom peak magnitude, an apparent extend in bloom duration, but no change in total chlorophyll during the bloom (also reported). This is just a quick idea and might be better captured by , which are mentioned in the introduction, but not used in the analysis. As I mentioned before, I think it is important to include the nutrient concentrations results in the manuscript to better support its conclusions. I would also suggest including the actual time series (environmental factors and fluorescence) as part of supplementary material.

*Authors' response: Unfortunately the temporal resolution of the nutrient concentration data is not sufficient to quantify the timing of nutrient uptake onset and light limitation alleviation – especially in winter when only few cruises are sampled for nutrients. Nevertheless, judging from the few transects sampled for nutrients in December and early January, nutrient limitation is alleviated well before light availability increases, and this has been our understanding of nutrient dynamics in the high latitude, semi-enclosed Baltic Sea. We looked into several metrics for nutrient uptake rates but could not link inter-annual nutrient variability to bloom phenology parameters. This may be due to the relatively sparse collection of bottle samples for laboratory analysis (on average every $6^{th}$ transect). While at present this result does not prompt further discussion in the manuscript, future studies may benefit from additional data so as suggested, we added nutrient metric results to the appendix. In addition we added the following to the manuscript:*
*'Several times ship service had not commenced early enough in the year to record bloom onset, which implies that trends in bloom start and nutrient peak timing could not be derived with the same accuracy and precision as the other phenological parameters. Nutrient metrics are provided in the appendix to aid future work, if additional data or longer time-series become available.'*

```
diff --git a/AlgalineSpringBloom.tex b/AlgalineSpringBloom.tex
index a8630da..e2e11b6 100644
--- a/AlgalineSpringBloom.tex
+++ b/AlgalineSpringBloom.tex
@@ -26,13 +26,14 @@

\Author[1, 2]{Philipp M. M.}{Groetsch}
\Author[3, 4]{Stefan G. H.}{Simis}
\Author[1]{Marieke\Author[5, 1]{Marieke A.}{Eleveld}
\Author[2]{Steef W. M.}{Peters}

\affil[1]{Institute for Environmental Studies (IVM). De Boelelaan 1087, 1081 HV
Amsterdam, The Netherlands}
\affil[2]{Water Insight. Marijkeweg 22, 6709 PG Wageningen, The Netherlands}
\affil[3]{Plymouth Marine Laboratory. Prospect Place, The Hoe, PL1 3DH Plymouth,
 United Kingdom}
\affil[4]{Finnish Environment Institute SYKE. Erik Palménin Aukio 1, 00560 Hels
inki, Finland}
\affil[5]{Deltares. P.O. Box 177, 2600 MH Delft, The Netherlands}

%% The [] brackets identify the author with the corresponding affiliation. 1, 2,
 3, etc. should be inserted.

@@ -58,11 +59,11 @@

\begin{abstract}
Phytoplankton spring bloom phenology was derived from a 15-year time-series (200
0-2014) of ship-of-opportunity chlorophyll-\textit{a} fluorescence observations
collected in the Baltic Sea.Sea through the Alg@line networ
k. Decadal trends were analysed against inter-annual variability in bloom tim
ing and intensity, and environmental drivers (nutrient concentration, temperatur
e, radiation level, wind speed).

Spring blooms developed along afrom the south to the nor
thgradient with the first blooms peaking mid-March in the Bay of Mecklen
burg and the latest bloom peaks occurring mid-April in the Gulf of Finla
nd. Bloom duration was similar between sea areas (\SI{43(2)}{\day}), except for
shorter bloom duration in the Bay of Mecklenburg (\SI{36(11)}{\day}). Variabilit
y in bloom timing increased towards the south. Bloom peak chlorophyll-\text
it{a} concentrations were highest (and most variable) in the Gulf of Finland
(\SI{20.2(57)}{\milli\gram\per\cubic\meter}) and the Bay of Mecklenburg (\SI{12.
3(52)}{\milli\gram\per\cubic\meter}).

Bloom peak chlorophyll-\textit{a} concentration showed a negative trend
of \SI{-0.31(10)}{\milli\gram\per\cubic\meter\per\year}. Trend-agnostic distribu
tion-based (Weibull-type) bloom metrics showed a positive trend in bloom duratio
n of \SI{1.04(20)}{\day\per\year}, which was not found forwith a
ny of the threshold-based metrics. The Weibull bloom metric results were conside
red representative in presence of bloom intensity trends.

Bloom intensity was mainly determined by winter nutrient concentration, while bl
oom timing and duration co-varied with meteorological conditions. Longer blooms
corresponded to higher water temperature, more intense solar radiation, and lowe
r wind speed. It is concluded that nutrient reduction efforts led to decreasing
bloom intensity, while changes in Baltic Sea environmental conditions associated
 with global change correspond to a lengthening spring bloom period.
\end{abstract}
@@ -74,13 +75,13 @@ Human influence and climate change transform terr
estrial and marine ecosystems w
```

Phytoplankton bloom intensity and timing (bloom phenology) are indicators for ecosystem health at the base of the food web \citep[e.g.][]{Hays2005,Adrian2009,Vargas2009}. Phenological studies are increasingly used to inspect regional ecosystem response to nutrient reduction efforts \citep{Helcom2007,Voss2011, Fleming-Lehtinen2015} and changing climatic conditions \citep{Sommer2008, Paerl2009}. The Baltic Sea is a coastal ecosystem affected by eutrophication \citep{Korpinen2012}, which intensifies naturally occurring spring- and summer bloom \citep{Bianchi2000, Helcom2007}. The Helsinki Commission formulated a nutrient reduction scheme aimed at improving ecosystem health in 1992 \citep{Helcom1992}, which entered into force in 2000. Monitoring of key ecosystem health indicators is implemented in the national monitoring programmes of HELCOM contracting parties. These programmes include traditional dedicated sampling campaigns at sea and increasingly the use of highly resolving observation platforms.

Ships-of-opportunity (typically cargo ships or passenger ferries) offer a largely weather-independent, reliable, and cost-effective platform for the collection of high frequency in situ observations \citep{Leppanen1995, Ainsworth2008}. Phytoplankton pigment fluorometers are included in most so called of the se ferryboxes. In the Baltic sea, such systems have recorded phytoplankton [31mbloom blooms on the route from Helsinki to Travemünde (v.v.) since 1992 \citep{Rantajarvi2003}. On this route route, ferryboxes have collected over 9.5 million chlorophyll-\textit{a} pigment fluorescence observations from 1926 transects with a median revisit time of under two days in the last 15 years (2000-2014). Ship-based observations from merchant vessels provide [31mcontinuity, continuity in monitoring, which is particularly important in seasons when other observation systems are less reliable. In spring, satellite observations are rare due to high average cloud cover, while high costs of dedicated research cruises and coastal laboratories limit their spatio temporal coverage. Ferrybox observations are therefore the primary source of observations to study spring bloom dynamics in this region.

Phytoplankton abundance and succession in the Baltic Sea is controlled by nutrient \citep{Neumann2002, Tamminen2007} and light availability \citep{Sverdrup1953, Smetacek1990,Nelson1991,Siegel2002}, mixing-status \citep{Ueyama2005a,Sharples2006}, temperature \citep{Grayek2011}, ice cover \citep{Kahru1990,Omstedt2004,Sommer2008}, and salinity \citep{Fennel1999,Tamminen2007}. In addition, the quantum yield of fluorescence is influenced by solar irradiance \citep{Kiefer1973,Dandonneau1997,Marra1997,Sackmann2008a}, species composition, and physiology \citep{Kiefer1989}. Hence, interpretation of unattended pigment fluorescence measurements in terms of phytoplankton biomass presents a number of challenges \citep{Roesler2013}. Firstly, phytoplankton distribution exhibits high spatial and temporal variability, while ferryboxes measure pigment fluorescence at fixed depth \citep{Ruokanen2003}. Therefore, stratified conditions may not be well represented in the data \citep{Groetsch2014}. Secondly, in a typical ferrybox setup fluorescence yield is at best determined as a daily sea area-average, re gional average, which disregards variability on smaller spatio-temporal scales. Despite these challenges, \citet{Fleming2006} demonstrated that ferrybox observations in the Baltic Sea can be used to derive bloom timing and intensity for biomass-rich sea areas. The authors reported They report a slightly negative trend in bloom initiation in the Northern Baltic Proper and the Gulf of Finland for the period 1992-2004. Recent studies also reported shifts in phytoplankton spring bloom biomass or species composition \citep[e.g.][]{Klais2011,Wasmund2011,Wasmund2013}, but \citep[e.g.][]{Klais2011,Wasmund2011,Wasmund2013}. \cite{Kahru2014} reported that the timing of cyanobacterial surface accumulations has advanced approximately 20 days from 1979 to 2013. However, information about                                                                              shifts in

Choosing an adequate bloom metric is not trivial as no strict clear [m guidelines exist that unambiguously recommend conclusively support one metric over the other. others. Bloom metrics for both remotely sensed and in situ sampled time series are commonly divided into three groups: 1) fixed or variable concentration threshold metrics \citep{Siegel2002,Fle

ming2006,Lips2014,Racault2015}, 2) growth-rate-based metrics \citep{Rolinski2007,Wiltshire2008}, and 3) distribution-based metrics \citep{Rolinski2007,Platt2009,Vargas2009,Zhai2011}. Threshold- and growth-rate based metrics typically require data pre-processing (e.g. interpolation and smoothing), to mitigate the impact of gaps, noise, outliers, and multi-modal bloom distributions toon the derived bloom phenology \citep{Rolinski2007,Cole2012,Ferreira2014}. Distribution-based metrics fit an analytical expression to observations using fitting routines designed to cope with imperfections in the input data while optimally preserving natural variability. Distribution-based bloom metrics are considered more robust than threshold- or growth-rate-based metrics, in the presence of complex, multi-modal bloom observations \citep{Ji2010}. Interpretation based on several, conceptually different bloom metrics can be used to obtain uncertainty estimates \citep{Ho2015}, and\citep{Ho2015}. It also allows to [31mqualitatively indicatescreen for long-term trends in bloom phenology. The latter is because threshold-based metrics are biased by long-term bloom intensity trends, whereas growth-rate and distribution-based metrics are not. Figure \ref{fig:trend_scheme} illustrates how a gradual decline (negative trend) in bloom peak concentration causes anythreshold-based metric based on fixed thresholds (e.g. derived from climatology or expert-judgement) to introduce an artificial negative trend in bloom duration. In contrast,growth-rate and distribution-based metrics based on growth rate, distribution, or annually derived thresholds yield a constantsingle bloom duration for the givenin this example because they are sensitive to concentration distributions, rather than absolute concentrations.bloom intensity does not influence these metrics.

The aims of this study are twofold: (1) to report long-term trends for Baltic Sea spring bloom intensity and timing, and (2) to attribute these trends to changes in environmental conditions. This paper describesTo meet these objectives, we describe a methodology to derive quality controlled time-series of chlorophyll-\textit{a} concentrations from observations collected by [32munder the Baltic Sea Alg@line programand its predecessors over a period of 15 years (2000-2014). Uncertainties arising from variability in the phytoplankton pigment fluorescence yield are estimated. Bloom phenology parameters based on several conceptually differingparameters, derived from threshold- and distribution-based bloom metricsmetrics, are[31mpresented, and explored for long-term trends. Inter-annual variability of bloom phenology parameters are attributed to nutrient availability and meteorological conditions (temperature, radiation level, wind speed), which might help to relate long-term trends to unique causes. Finally, we summarize how these results contribute to the discussion on recent changes in the Baltic Sea, and the monitoring practices that need to be in place to detect such changes.

%%%%%%%%%%%%%%%%%%%%%%%%%%%%%%%%%%%%%%%%%%%%%%%%%%%%%%%%%%%%%%%%%%%%%%%%
\section{Materials and Methods}
@@ -89,13 +90,13 @@ The aims of this study are twofold: (1) to report long-term trends for Baltic Se
\subsection{Alg@line Data}
\label{ssec:algaline}

In situ data in this study were collected until 2009 by the Finnish Institute of Marine Research, and by the Finnish Environment Institute (SYKE) from 2009 onwards, within the Alg@line network of Baltic Sea ferryboxes. Here we consider systems installed on two cargo vessels, M/S \emph{Finnpartner} (2000-2006) and M/S \emph{Finnmaid} (2007-2014), which served between Travemünde (Germany) and Helsinki (Finland) as depicted in Fig. \ref{fig:transect}. Three routes were sailed during the study period. Depending on wave height and directionweather conditions the passage between Gotland and the mainland of Sweden (52 \%)(39 \% of all transects) was favoured over the direct route east of Gotland (39(52 \%), while the route with a lay-over in Gdansk (Poland) was only occasionally served during 2009 to 2012 (7 \%). Several transects (2 \%) were sailed for refuelling or maintenance in other ports and not use

d for this study.

Details on the instrumentation of the Alg@line ferrybox systems can be found in \citet{Leppanen1994, Rantajarvi2003, Ruokanen2003, Seppala2007}. In summary, the systems recordedrecord in vivo fluorescence of chlorophyll-\tex tit{a} (chla), salinity and temperature throughout the studied period (2000-2014). Turbidity and (in summer) phycocyanin pigment fluorescence were recorded from 2005 onwards and are not used here. At cruising speed (\SIrange{20}{23}{\knot}) the sampling interval of \SI{20}{\second} resulted in a nominal spatial resolution of \SI{200}{\meter}.

Quality control flags were defineddrived from (1) sensor reading thresholds on speed, flow rate, hull and line temperatures,sampled water temperature, and (2) data variability, expressed as lower and upper bounds for standard deviation between neighbouring measurements. [32mmeasurements, as described below. Measurements at low ($<$ \SI{5}{\knot}) or zero ship speed are typically collected in harbour and were disregarded.omitted. Erroneous records, e.g. caused by instrument communication errors, were removed using a moving window mean filter. A window length of 25 observations (approximately \SI{8.3}{\minute}) was used for records of ship speed, and a window length of 100 observations (\SI{33.3}{\minute}) was used for flow rate and temperature records. Low flow rates can indicate blocked passages, pump failure, or leaks. Flow meter readings were available for approximately one-third of all records. A proxy for flow disruption is the difference in ship-hull temperature and in-line temperature. Flow rates $<$ \SI{0.3}{\liter\per\minute} or a temperature difference $>$ \SI{2}{\degreeCelsius} were used to flag records as suspect. Instrument failure, communication and digitizing errors may lead to 'stuck' values, which were detected by calculating standard deviation in a moving window of 100 samples. Observations corresponding to low standard deviation ($\sigma<1e^{-4}$) of chla fluorescence measurements or GPS-derived latitude were omitted. GPS-derived latitude was additionally filtered for exceptionally high short-term variability ($\sigma>0.5$, window size 50 samples), caused by poor satellite reception or serial communication errors. Table \ref{tab:qc} provides an overview of the applied quality control flags.

Chla fluorescence data were corrected for sensor drift and discontinuities by transect-wise normalization (division by transect mean). This was necessary to account for changes in instrumentation, signal contamination due to bio-fouling, trapped bubbles and particles, and changes in sensor sensitivity due to deterioration or manual adjustments.Laboratoryadjustments. Laboratory analysis results of bottle samples are typically available from every 6$^{th}$ transect, with up to 24 samples collected by automated, refrigerated water samplers (Teledyne Isco). Laboratory analyses included inorganic nutrient concentrations (nitrate+nitrite, phosphate and silicate), chla concentration, and occasionally inverted light microscopy counts of phytoplankton species. Laboratory chla concentration results were used to convert transect-normalized chla fluorescence to units of chla concentration (in \si{\milli\gram\per\cubic\meter}). First, a linear (generalized least squares) regression fit of normalized chla fluorescence against corresponding chla lab measurements was carried out for each sampled transect. If the regression failed ($R^2 < 0.3$ or $p > 1$) a moving window regression was carried out (window length 10 samples) and the subset with the highest $R^2$ was used to determine the correction factor. The threshold for $R^2$ was determined manually based on the distribution of $R^2$, while $p>1$ indicates numerical instabilities during the fitting procedure. Each transect without corresponding bottle samples was corrected by individually applying the regression parameters of the two neighbouring sampled transects. These two solutions were then interpolated linearly, weighted by their temporal distance to the respective transect. Negative concentration values occasionally occurred for weak fluorescence signals, and were set to zero.

The diurnal variability of the fluorescence signal was estimated from quality-controlled observations in all seasons. First, these observations were divided by

their respective transect mean to remove biomass-driven first-order variability in the fluorescence signal. Then, diurnal cycles were derived by dividing these observations into hourly bins and sun elevation angle ranges (0.1 rad bins).

@@ -103,24 +104,24 @@ The diurnal variability of the fluorescence sig
nal was estimated from quality-co
Photosynthetically active radiation (\textsc{par}), sea surface temperature (\textsc{sst}) and wind speed (\textsc{wind}) were derived from the ECMWF ERA-Interim reanalysis data set \citep{Dee2011}. The spatial resolution of the model is constrained by the underlying atmospheric model, which is stored on a spatial T255 grid corresponding to approximately \SI{79}{\kilo\meter} cell size when projected to a reduced Gaussian grid. Four values per day were retrieved for each parameter and the entire Baltic Sea. Parameter values for each Alg@line observation were extracted using spatio-temporal spline interpolation of third order. The first order seasonal signal (e.g. rising \textsc{par} and \textsc{sst} in spring) was removed from the observations by subtracting multi-year (2000-2014) daily sea area averages, approximated by second order polynomials. The seasonally detrended parameters were then averaged over the bloom period and are further referred to as \textsc{par}, \textsc{sst}, and \textsc{wind}.

\subsection{Nutrient Concentration and Depletion Timing}
A single term for nutrient availability was adopted from \cite{Fleming2006}, calculated as $\textsc{nut} = \sqrt[3]{(NO_{3}+NO_{2}) \times PO_{4} \times SiO_{4}}$, where $NO_{3}+NO_{2}$, $PO_{4}$ and $SiO_{4}$ are the concentrations of nitrite+nitrate, phosphate, and silicate, respectively. These concentrations were derived from laboratory analysis of bottle samples that were regularly collected along the transect (further detail in section \ref{ssec:algaline}).          \textsc{nut} was spatially binned for each investigated sea area and re-sampled to daily averages and consecutively smoothed with a 21-day centred-running-mean filter. This treatment resembles theAlg@aline processing of Alg@aline observations (see section \ref{ssec:bloom_timing}) to enable consistent interpretation of the joint data set. Nutrient concentrations and depletion timing are described using the following metrics. The nutrient concentration prior to bloom start (\textsc{nut-peakvalue}) was defined as the yearly maximum nutrient concentration (day-of-year between 31 and 160). The day-of-year when the nutrient concentrations equalled 100 \%, 50 \%, and 25 \% of their peak values are referred as \textsc{nut-peakday}, \textsc{nut-deplday-50}, and \textsc{nut-deplday-25}. The day and value of the lowest nutrient concentration index are referred to as \textsc{nut-minday} and \textsc{nut-minvalue}. The rate of nutrient depletion between 75 \% and 25 \% of the peak value (\textsc{nut-slope}) was determined through linear regression.

\subsection{Extraction of Bloom Timing and Intensity}
\label{ssec:bloom_timing}
Extraction of bloom timing and intensity was carried out for five Baltic Sea areas, where each area follows definitions of the HELCOM Combine program \citep{Helcom2013}. Figure \ref{fig:transect} illustrates the location of the areas: the Western Gulf of Finland (\textsc{gof}: $>$59.5 \si{\degree N} latitude, along-transect), the Northern Baltic Proper (\textsc{nbp}: 58.4-59.5 \si{\degree N} latitude, along-transect), the combined Western and Eastern Gotland basins (\textsc{got}: 56.2-58.4 \si{\degree N} latitude, along-transect), the Southern Baltic Proper (\textsc{sbp}: 54.5-56.2 \si{\degree N} latitude, along-transect), and the Bay of Mecklenburg (\textsc{bom}: $<$54.5 \si{\degree N} latitude, along-transect). For the \textsc{got} and \textsc{sbp} areas only routes that passed by Gotland were selected whereas routes via Gdansk were excluded. This is because the route through Gdansk was sailed only from 2009 to 2012. If not otherwise stated, all further steps are carried out individually for each of these areas and for day-of-year between 31 (1(31 January) and 160 (9 June). The ship-of-oppportunity (Alg@line) measurements typically commenced in the second half of January, which is why 31 January was chosen as the start of our analysis. The end date was chosen such that it covers all spring bloom events in all basins but excludes summer bloom.

Alg@line chla concentrations (see section \ref{ssec:algaline}) were resampled to daily sea area averages, using linear interpolation, and subsequently smoothed with a 21-day centred-running-mean filter \citep[e.g.][]{Ferreira2014,Racault2015} to fill in gaps and reduce short-term variability. We derive several metrics, all of which have in common that the bloom peak concentration (\textsc{peakheight}, see Table \ref{tab:params} for explanations of acronyms) and timing (\textsc{peakday}) are defined as the maximum chla value at the corresponding day-of-year, respectively. Two threshold-based metrics and one distribution-fit-based metric were calculated:

1) Chla concentration exceeding a fixed-threshold of \SI{5}{\milli\gram\per\cubic\meter} was defined as bloom by \citet{Fleming2006}, further referred to as \texttt{const5}. A 21-day centred-running-mean filter was used to keep results comparable to the other metrics (\citet{Fleming2006}:considered
, whereas \citet{Fleming2006} used a 7-day centred-running-median filter
).filter.

2) \citet{Siegel2002} proposed a variable-threshold metric based on the 5 \%-above-median concentration, but reported small quantitative differences for thresholds between 1 and 30 \%-above-median. Their threshold is based on the complete annual cycle, while here only the spring bloom period from day-of-year 31 to 160 is considered. We refer to this metric as \texttt{median5}.

3) Distributions proposed to describe bloom phenology include shifted-Gaussian \citep{Platt2009}, Gamma \citep{Vargas2009}, and Weibull distributions \citep{Rolinski2007}. While theThe shifted Gaussian is symmetric in shape,
 whereas Gamma distributions allow for different slopes of bloom rise and decline. In addition, Weibull functions recognize non-zero offsets before and after the bloom phase. The latter has proven essential to obtain a good fit for the transition phase between spring and summer bloom.bloom with the
 here analysed data set. A modified Weibull-function, as proposed by \cite{Rolinski2007}, was fitted non-linearly to the preprocessed and scaled (to a range of zero to one) chla concentrations. The bloom initiation and end are defined as the $10^{th}$ and $90^{th}$ percentiles before and after the bloom peak, respectively. This metric is further referred to as \texttt{weibull}.

For each metric, bloom initiation, peak, and end dates (\textsc{startday}, \textsc{peakday}, and \textsc{endday}) were extracted from the data set. Based on these dates, bloom duration (\textsc{duration}), concentration average (\textsc{concavg}), and the sum of daily chla concentrations (\textsc{bloomidx}) were calculated. The latter was proposed by \citet{Fleming2006} to characterize bloom intensity. We assumed the bloom to have started prior to Alg@line service commence if the first data point already satisfied the bloom criterion for a given metric. Such cases were identified for 30 out of 225 combinations of sea region, year, and bloom metric (9 times for bloom metric \texttt{const5}, 16 times for \texttt{median5}, and 5 times for \texttt{weibull}). Corresponding bloom start days were replaced by the median value for the region over the 15 years studied in all subsequent calculations.

\subsection{Principal Component Analysis}
Principal component analysis (PCA) was carried out to attribute seasonally detrended meteorological conditions (\textsc{sst}, \textsc{par}, \textsc{wind}) and nutrient concentrations (\textsc{nut-peakvalue}, \textsc{nut-minvalue}) to the inter-annual variability in bloom intensity (\textsc{bloomidx}, \textsc{concavg}, \textsc{peakheight}) and timing (\textsc{startday} and \textsc{peakday}, \textsc{duration}). Outliers were defined for each parameter as departure by more than 3 standard deviations from the parameter mean, and replaced with the region-median. Z-score normalization (subtraction of mean, division by standard deviation) was carried out on a per-region basis.For 30 out of 225 combinations of sea

region, year, and bloom metric, ferrybox records started after blooms had initiated. Such cases were identified 9 times for bloom metric \texttt{const5}, 15 times for \texttt{median5}, and 5 times for \texttt{weibull}. Corresponding bloom start days were replaced by the median value for the region over the 15 years studied during subsequent calculation of bloom phenology trends. Region-equed, zero-mean and unit-variance data were then subjected to the PCA function in the python framework scikit-learn \citep{Pedregosa2011}.

%
%%%%%%%%%%%%%%%%%%%%%%%%%%%%%%%%%%%%%%%%%%%%%%%%%%%%%%%%%%%%%%%%%%%%%%%%%%%
@@ -131,17 +132,17 @@ Principal component analysis (PCA) was carried out to attribute seasonally detre

The Alg@line ferrybox systems collected over $9.5\times10^{6}$ observations between 2000 and 2014, of which $3.8\times10^{6}$ observations were sampled during spring (day-of-year 31 to 160). Availability and rejection rates for each quality control parameter are listed in Table \ref{tab:qc}. In total, quality control procedures removed 4.55 \% of all observations.

Determination of the fluorescence yield was supported by an 'adaptive regression' method. Where necessary ($R^2 < 0.3$ or $p > 1$), it selected the subset of bottle-sampled and laboratory-analysed chla concentrations that yielded the best linear fit to chla fluorescence observations.observations for a given transect. This procedure allowed to successfully fit 318 (98 \%) out[m ofthe total 324 transects withfor which bottle samples.samples were collected. Only 266 (82 \%) transects could have been used ($R^2 >= 0.3$ and $p \ll 1$) without applying this technique.

Figure \ref{fig:diurnal_variability}A shows normalized fluorescence observations as a function of sampling time-of-day. Results are presented separately for summer (May to August), winter (November to February) and the transition periods (autumn, spring). Diurnal variability was most pronounced in summer, when the fluorescence signal varied on average 50 \% over the course of a day. In winter and during the transition periods (spring, autumn)the effect was less pronounced, although a diurnal variability of 35 and 38 \% is still\%, respectively, was contained in therespective fluorescence signals. This seasonal effect is likely caused by variations in average irradiance intensity, which are modulated primarily by sun elevation, but also by atmospheric conditions (e.g. cloud cover, aerosol optical thickness) and optical properties of the water body (e.g. ice cover, attenuation). Figure \ref{fig:diurnal_variability}B depicts normalized fluorescence as a function of solar elevation. In this representation seasonal differences in diurnal variability are essentially absent and the correspondence between solar elevation and average fluorescence response was approximately linear for daytime observations.

\subsection{Bloom Intensity and Timing}
Blooms generally developed first in the south and progressed towards the north (see Fig. \ref{fig:phenology_geo_timing} and Table \ref{tab:bloomstats}). Bloom peak timing (not influenced by choice of metric) followed this pattern, as did metric-dependent bloom start and end dates. The fixed-threshold bloom metric \texttt{const5} suggested longer blooms in high-biomass sea areas like the \textsc{gof}, compared to low-biomass areas such as the \textsc{sbs}. The variable-threshold metric \texttt{median5} applies area-specific bloom thresholds (\textsc{nbp}: \SI{3.52}{\milli\gram\per\cubic\meter}, \textsc{gof}: \SI{4.95}{\milli\gram\per\cubic\meter}, \textsc{got}: \SI{2.51}{\milli\gram\per\cubic\meter}, \textsc{sbs}: \SI{2.62}{\milli\gram\per\cubic\meter}, \textsc{bom}: \SI{4.02}{\milli\gram\per\cubic\meter}) and resulted in approximately stable bloom durations for[mduration in all sea areas. The \texttt{weibull} metric, which is not sensitive to absolute bloom intensity, also resulted in comparable bloom durations for all sea areas. The year-to-year variability of start, peak, and end days generally increased towards the south for all metrics.

Spring bloom intensity was described by three parameters: the metric-independent bloom peak concentration (\textsc{peakheight}), the chla concentration average during bloom conditions (\textsc{concavg}), and the sum of daily chla concentrations over the bloom period (\textsc{bloomidx}). Similar patterns were observed for all these parameters and bloom metrics, as illustrated in Fig. \ref{fig:phenology_geo_intensity}. The highest bloom intensity was found in the \textsc{gof} and \textsc{nbp}, followed by the \textsc{bom}. Low-intensity blooms were observed in the \textsc{sbp} and the \textsc{got}. Variability was generally proportional to bloom intensity, highest in the high-biomass and coastal \textsc{gof} and \textsc{bom}. Variability in \textsc{bloomidx} was comparable to that in \textsc{peakheight}, while \textsc{concavg} was considerably more stable. All calculated bloom phenology parameters can be found in the supplementary material.

\subsection{Trends}
Figure \ref{fig:trends} shows normalized (subtraction of area-average concentration) \textsc{concavg} and \textsc{peakheight} for all sea areas combined, as a function of bloom year. \textsc{peakheight} is independent of bloom metric and shows a highly significant ($R^2 = 0.12$, $p \ll 0.01$) negative trend of \SI{-0.30(10)}{\milli\gram\per\cubic\meter\per\year}. \textsc{concavg} is dependent on bloom start and end days and was therefore calculated for all applied metrics. Statistically significant, negative trends resulted from all metrics: \SI{-0.12(4)}{\milli\gram\per\cubic\meter\per\year} for \texttt{const5} ($R^2 = 0.11$, $p \ll 0.01$), \SI{-0.11(5)}{\milli\gram\per\cubic\meter\per\year} for \texttt{median5} ($R^2 = 0.12$, $p < 0.05$), and \SI{-0.22(7)}{\milli\gram\per\cubic\meter\per\year} for \texttt{weibull} ($R^2 = 0.11$, $p \ll 0.01$).

No significant trends were found for \textsc{bloomidx}, \textsc{startday}, and \textsc{peakday} with any of the applied metrics, while \textsc{endday} showed weakly correlated but statistically significant ($R^2 = 0.06, 0.08, p<0.05$) positive trends for \texttt{const5} and \texttt{weibull} with slopes \SI{0.6}, \SI{0.7(3)}{\day\per\year}, respectively.
@@ -151,11 +152,11 @@ Bloom duration resulting from the \texttt{weibull} metric stands out in the resu
Peak nutrient concentrations showed no significant trend, in contrast to post-bloom nutrient concentrations with a highly significant, negative trend \SI{-0.020(4)}{\micro\mol\per\liter\per\year} ($R^2=0.23$, $p \ll 0.01$). Peak nutrient concentration timing shifted to earlier dates (\SI{-0.7(3)}{\day\per\year} ($R^2=0.06$, $p<0.05$)), while the 25 \%-of-peak-value was reached progressively later (\SI{0.67(31)}{\day\per\year}, ($R^2=0.06$, $p<0.05$)). No significant trends were found for nutrient depletion slope, 50 \%-of-peak-value-timing, or day of minimal nutrient concentrations.

\subsection{Inter-annual Variability}
Pre-bloom nutrient concentrations were positively correlated to bloom peak height (no normalization, $R^2=0.39$, $p \ll 0.01$) and concentration average (no normalization, $R^2=0.37 - 0.57$, $p \ll 0.01$, depending on metric). After applying area-wise mean and variance (z-score) normalization, however, a negative correlation was found for \textsc{peakheight} ($R^2=0.11$, $p \ll 0.01$, metric independent) and \textsc{concavg} ($R^2=0.12, 0.11$, $p \ll 0.01$ for \texttt{const5} and \texttt{weibull}, respectively).

The timing of nutrient depletion, specifically \textsc{nut-deplday-50}, was positively correlated to the bloom peak day ($R^2=0.47$, $p \ll 0.01$), and to bloom-averaged, detrended \textsc{par}-levels ($R^2=0.14-0.29$, $p \ll 0.01$). Average wind speed and \textsc{par} were negatively correlated during bloom conditions ($R^2=0.10-0.23$, $p \ll 0.01$). The bloom timing parameters (\textsc{startday}, \textsc{peakday}, \textsc{endday}) were weakly but statistically significantly inter-correlated (results not shown).

PCA scores and loadings of the first three principal components (PC) are shown a

s biplots in Fig. \ref{fig:pca_biplots}. The first PC is dominated by negative c
orrelations to bloom intensity parameters (\textsc{peakheight}, \textsc{concavg}
, \textsc{bloomidx}). This component is positively correlated to pre-bloom nutri
ent concentration (\textsc{nut-peakvalue}) and bloom duration, illustrating that
 bloom intensity is affecteddriven by pre-bloom nutrient availab
ility. The second PC is linked to bloom timing, with strong positive correlation
s to \textsc{startday} and \textsc{peakday}. Correlations to \textsc{par} (posit
ive), \textsc{sst} (positive), and \textsc{wind} (negative) suggest that weather
 conditions affect bloom timing. Bloom duration is positively correlated to the
third PC, as well as to \textsc{bloomidx}. Additional negative correlations to \
textsc{nut-minvalue} and \textsc{wind}, as well as a positive correlation to \te
xtsc{par}, suggest a link between favourable meteorological conditions (low wind
-mixing, high light level) and efficient nutrient depletion.
%
%%%%%%%%%%%%%%%%%%%%%%%%%%%%%%%%%%%%%%%%%%%%%%%%%%%%%%%%%%%%%%%%%%%%%%%%%%%%%%%%

@@ -165,37 +166,40 @@ PCA scores and loadings of the first three prin
cipal components (PC) are shown a
Trends in spring bloom phenology can be interpreted as responses to nutrient red
uction as well as to slowly acting environmental processes, such as climate chan
ge. To disentangle or even quantify these trends, suitable observation platforms
 and subsequent analytical approaches must be chosen. We present evidence that f
undamental challenges of ferrybox observations can be overcome to yield an inter
nally consistent data source. Subsequently, the behaviour of commonly used bloom
 metrics in presence of decadal trends can be scrutinized in the context of prev
iously reported system knowledge. Finally, we attempt to disentangle the effects
 of nutrient availability and meteorological conditions on inter-annual variabil
ity in bloom phenology.

\subsection{Automated Processing of Ferrybox Observations}
Thresholds for speed, flow rate, and data variability were iteratively adjusted
to the data set and might thereforemay not apply directly[
mbe applicable to other ferrybox implementations. Particularly flow [31
mraterate, derived from differences in line and hull temperature [32
mwill likely requiresrequire tuning to each ferrybox installa
tion. However, here we analysed data from two ferrybox installations, which
 could be treated with the same set of thresholds. Transect-wise normalizatio
n of the quality controlled fluorescence data was adequate to consistently inter
pret observations collected by different generations of instrumentation. However
, this approach crucially depends on continuous temporal coverage of reference m
easurements for calibration to chla concentrations. Adaptive regression analysis
 improved the handling of statistical outliers which would otherwise hamper dete
rmination of fluorescence yield, while transects for which no bottle samples are
 available were corrected with an interpolated fluorescence yield derived from t
he closest bottle-sampled transects. The present procedure allows for automated
and reproducible processing which is an improvement over manual quality control.
 Applying the proposed interpolated fluorescence yield helps in reprocessing and
 long-term data analysis of ferrybox fluorescence observations to better represe
nt natural variability.

\subsection{Variability in Fluorescence Yield}
Fluorescence diurnal variabilityDiurnal fluorescence patterns sh
owed low seasonal dependence after accounting for solar elevation. Unsurprisingl
y, light intensity is the predominant factor in Baltic Sea phytoplankton fluores
cence yield variability. Other seasonal differences in fluorescence response can
 be attributed to typically higher cloud cover in winter compared to summer and
spring/autumn, which was not accounted for in our analysis. The seasonal cycle o
f species composition, from dinoflagelate and diatom dominated spring communitie
s \citep{Klais2011} to cyanobacterial summer bloom \citep{Kahru2014}, influenced
 fluorescence yield considerably less than diel cycles.

The diurnal variability in fluorescence response of 50 \% during an average summ

er day is within athe range of earlier findings, e.g. 66 \% ($\p
m 33 \%$) for near surface observations in upwelled waters of the equatorial Pac
ific reported by \cite{Dandonneau1997} or 30 \% for near-surface seaglider obser
vations in Northeast Pacific waters off the Washington coast, USA \citep{Sackman
n2008a}, although differences in normalization impede direct comparison. The sam
pling depth of \SI{5}{\meter} for Alg@line systems and the high attenuation of t
he Baltic Sea in comparison to clear Pacific Ocean waters cause lower[3
2mare likely to dampen the observed diurnal variability.

In this studystudy, fluorescence observations during spring, whe
n diurnal variability reached on average 38 \%, were binned for five large Balti
c Sea areas. At a typical cruising speed of approximately \SI{23}{\knot} each se
a area is sampled for at least several hours. This limits the influence of diurn
al variability in fluorescence yield along a transect,transect on d
erived chla concentration, which is therefore of lesser relevance for the pre
sent study. However, if fluorescence measurements were to be quantitatively eval
uated at a higher spatial resolution, variablelocally varying fl
uorescence yield should be accounted for. Analysis of signal-coherence \citep{Gr
oetsch2014} offers an alternative to quantitative interpretation of fluorescence
 observations and can be used to qualitatively detect cyanobacterial surface blo
om. If light history is known, e.g. from a dedicated irradiance sensor, a correc
tion of diurnal fluorescence yield variability might be possible and further res
earch in this direction is recommended.

\subsection{Spring Bloom Timing and Intensity}
The presented bloom phenology expands the time series presented by \cite{Fleming
2006} and is in good agreement for the overlapping period (2000 - 2004) when com
paring the \texttt{const5} metric results. Remaining differences are likely due
to quality-control and pre-processing procedures on the fluorescence records. [
31mIn their work, theThe authors reported for \textsc{gof}, \textsc{n
bp}, and the Arkona Sea that bloom typically started in the south and ended in t
he north, while bloom intensity increased towards the north. These observations
are confirmed here. Sea areas not covered in \cite{Fleming2006}, e.g the high-bi
omass \textsc{bom} and low-biomass \textsc{sbp} and \textsc{got}, followed the r
eported south-north trend in bloom development. Present results also support and
 expand the findings of \cite{Fennel1999}, who showed with simulations and monit
oring data from 1994-1996 for the Western Baltic Sea that surface heating in ear
ly spring needs to overcome the temperature of maximum density to repress convec
tive mixing and allow spring bloom to emerge. The temperature of maximum density
 increases with decreasing salinity, so that convective mixing is sustained long
er in less saline northern Baltic Sea waters when spring temperature is on the r
ise. At the same time, incident solar radiation increases slower in the north du
e to lower solar elevation.

\subsection{Trends}
Interannual variability in coastal systems exceeds long-term trends by orders of
 magnitude \citep{Cole2012}. Consequently, trends were observed at relatively lo
w coefficients of correlation. The importance of appropriate data pre-proce
ssingpreprocessing and gap-handlinggap handling \cite
p[e.g][]{Cole2012,Racault2014a} and choice of metric \citep{Ferreira2014} has be
en emphasizeddemonstrated in literature and is further demo
nstratedemphasized by the present analysis. Robustness of the reporte
d decadal trends is documented by high statistical significance levels ($p\ll0.0
1$, Figs. \ref{fig:trend_duration} and \ref{fig:trends}), which were supported b
y spatially binning phenology parameters from all examined Baltic Sea areas. Sim
ilar trends were observed earlier for individual Baltic Sea areas, however, usua
lly outside 95 \% confidence intervals \citep[e.g][]{Wasmund2003}.

\cite{Helcom2014} reported stable or increasing chla concentrations for the peri
od 2007-2011 in several Baltic Sea areas despite signs of declining nutrient con
centrations. More recently, eutrophication trend reversal and oligotrophication
processes were reported by \cite{Andersen2015}, based on analysis of 112 years o

f consolidated Baltic Sea observations. Both reports considered surface-layer ch
la concentration in summer as one of the direct indicators for eutrophication, b
ut did not include spring bloom in their assessment. The time series for 2000-20
14 that we present here fills this gap: a negative trend in bloom intensity was
[31mfound also found for spring bloom, providing further evidence for
 their hypothesis.hypothesis of gradual nutrient load reduction.

The concentration distribution-ratios on which the Weibull-metric is based
are calculated for each bloom individually, in contrast to the thresholds[32
mThresholds of \texttt{const5} and \texttt{median5}that are fixed for
 the completewhole time series (see Fig.series. The
 observed negative trend in peak concentration introduces an artificial negative
 trend in bloom duration because an increasingly higher percentile of the distri
bution is seen below the bloom threshold (Fig.                          \ref{fig:t
Threshold-based metricsContrary to this expected behaviour, however, \te
xttt{const5} and \texttt{median5} revealed no significant trends in bloom [3
1mduration, whileduration. This indicates that the anticipated negative
trend in bloom duration was countered by a positive trend, e.g. in bloom intensi
ty. The Weibull-metric is based on concentration distribution-ratios that are ca
lculated individually for each bloom. Therefore, Weibull-metric results for bloo
m duration are not sensitive to long-term trends in peak concentration.          Weibu
ll-distribution metrics showedconfirmed a highly significant, po
sitive trend.trend in bloom duration. These two contrasting
sets of results nevertheless supportcorroborate the c
onclusion that spring blooms in the Baltic Sea have become longer, while chla pe
ak and average concentration levels declined.

This 'flattening' of the concentration distribution is supported by the absence
of a trend in time-integrated biomass \textsc{bloomidx} and by shifts in nutrien
t concentration timing (earlier nutrient peak concentration, later 25 \%-of-peak
-value day). These results indicate that annually generated spring bloom biomass
 has not changed significantly over the study period, in contrast to bloom timin
g. \cite{Kahru2014} found a similar development for cyanobacterial summer surfac
e bloom, and reported decadal oscillations, yet no long-term trend, of surface a
rea covered by cyanobacteria in the period 1979-2013. In the same period, summer
 bloom initiation moved to earlier dates by \SI{-0.6}{\day\per\year}. These resu
lts suggest that the gap has decreased between dinoflagelate- and diatom-dominat
ed spring bloom and cyanobacterial summer bloom. Due to the shorter period
covered here as compared to the time series presented by \cite{Kahru2014}, it ca
nnot be ruled out that the spring bloom trends are caused by decadal oscillation
. Moreover, Alg@line nutrient records often did not commence sufficiently early
in the season to record bloom onset. Trends in bloom start and nutrient peak tim
ing can therefore not be derived at the same accuracy and precision as the other
 phenological parameters. In future, additional data and longer time series may
revise this analysis. To this end, nutrient metrics derived in this work are pro
vided in the appendix.

Our findings emphasize that bloom timing is an essential indicator to monit
or marine ecosystem dynamics, and thus eutrophication status. Observations at hi
gh temporal resolution and choice of bloom metrics are crucial to derive bloom t
iming trends. Eutrophication status assessment frameworks such as HEAT3.0 \citep
{Andersen2015} may be adapted to embrace available high-frequency data sources t
o include bloom timing in their analysis. The present results may also prove use
ful in the calibration and validation of ecosystem models of the Baltic Sea.

Our findings emphasize that bloom timing is an essential indicator to monit

or marine ecosystem dynamics, and thus eutrophication status. Crucial for deriving bloom timing trends are observations at high temporal resolution and choice of bloom metrics. Eutrophication status assessment frameworks such as HEAT3.0 \citep{Andersen2015} may be adapted to embrace available high-frequency data sources to include bloom timing in their analysis. Ecosystem models of the Baltic Sea and other coastal or inland systems can also use the presented results for validation and to enhance their predictive capabilities.

\subsection{Environmental Forcing}
Gradually decreasing nutrient concentrations \citep{Helcom2014, Andersen2015}, as well as rising average air- and sea-surface temperatures \citep{Omstedt2004, Helcom2013c} have been reported for recent years, corresponding to a combination of nutrient-reductionnutrient reduction efforts and global climate change. Several scenarios for future change are plausible \citep{Duarte2009} but extrapolation of the present results to climate scenarios is beyond the scope of this study. However, weWe nevertheless make an attempt to attribute the observed bloom phenology shifts to reported changes in environmental drivers.

Winter-time nutrient concentration and bloom intensity were positively correlated if no spatial normalization was applied. This supports the paradigm that the first-order driver of bloom intensity is nutrient availability. Therefore, lacking otherLacking alternative explanations, we attribute the reported negative trend in bloom peak concentration to declining nutrient concentrations. First-order spatial trends in bloom intensity and timing can be removed by an area-wise z-score normalization, which effectively constrains the analysis to inter-annual variability. After this normalization both regression and PCA resulted in negative correlation between winter-time nutrient concentration and bloom intensity. This negative feedback can be understood as a subtle interaction between meteorological forcing and nutrient supply: strong wind-forced mixing can cause upwelling of deep, nutrient rich waters to surface layers. Wind speed, however, was found to be negatively correlated to the prevalent light level, as well as to bloom duration and bloom index. Therefore, in years when additional nutrients are available due to strong wind forced mixing, low-light regimes that can hamperslow down bloom development are also likely to prevail.

Bloom duration co-varied primarily with weather conditions, e.g. high irradiance levels and low wind speeds were frequently observed for long-lasting blooms (and vice versa). Although the same pattern was observed for bloom timing, no trend was found for bloom start- and peak-day. Increasingly favourable meteorological conditions in late bloom phases are thus a likely driver for the observed increase in bloom duration. Similar weather-driven modulations of bloom timing were reported earlier \citep{Fleming2006,Meier2011,Neumann2012} for spring, and especially cyanobacterial summer bloom \citep{Wasmund1997,Kanoshina2003,Wynne2010,Wynne2011}.

\conclusions
\label{sec:conclusions}
A Baltic Sea spring bloom phenology was derived from 15 years of automated ferry box chla fluorescence observations. Procedures for automated quality-controlquality control and processing were introduced and uncertainty due to diurnal variability in phytoplankton fluorescence response was quantified.resolved. Both innovations promote increased use of ferrybox observations for scientific research and monitoring purposes, such as the periodic HELCOM eutrophication status assessments. Negative trends in spring bloom peak- and average-concentration were found and an increase in bloom duration was derived from conceptually differing bloom metrics. Inter-annualInter annual variability in bloom intensity was primarily linked to nutrient availability, while bloom timing and duration was found to be related to meteorological conditions. In the future, these findings might allowhelp to better disentangle ecosystem response to changing nutrient availability and clim

atic conditions.

%\appendix
%\section{}    %% Appendix A
@@ -204,7 +208,7 @@ A Baltic Sea spring bloom phenology was derived f
rom 15 years of automated ferry

\begin{acknowledgements}
The authors thank the Alg@line consortium, specifically scientists and technical
 personnel at SYKE (and formerly FIMR),and FIMR, for the ferrybo
x in situ data set. Acknowledgement is made to ECMWF for the use of their ERA-In
terim data set in this research. MAE, SWMP and PMMG were co-funded by the Europe
an Community Seventh Framework Programme under grant agreement 607325 AQUA-USERS
, and grant agreement 313256 GLaSS. PMMG also received support from EC/IAPP proj
ect WaterS (Grant 251527). We sincerely thank the anonymous reviewers for t
heir detailed comments and constructive criticism on the manuscript.
\end{acknowledgements}

diff --git a/AlgalineSpringBloom_figures_tables.tex b/AlgalineSpringBloom_fi
gures_tables.tex
index aa1063f..62a9102 100644
--- a/AlgalineSpringBloom_figures_tables.tex
+++ b/AlgalineSpringBloom_figures_tables.tex
@@ -25,7 +25,7 @@
% Table with QC
%
\begin{table}[t]
\caption{Quality control flag definitions and statistics. Observations were [31
mexcludedomitted if any of the flags exceeded the respective threshol
d. Absolute temperature difference is measured between the water intake and the
flow-through sensors. Availability and rejectrejection rates wer
e calculated relative to the total number of data points.observatio
ns. }
\begin{tabular}{l||rrrrr}
& \textbf{Sign} & \textbf{Threshold} & \textbf{Availability [\%]} & \textbf{Reje
ction Rate [\%]} \\ \hline
Speed, [\si{\knot}] & $<$ & 5 & 100 & 1.33 \\
@@ -124,7 +124,7 @@ All & & & & 4.55 \\
\begin{figure}[t]
  \includegraphics[width=8.3cm]{/home/phil/Documents/work/dev/AlgalinePaper/spri
ng_track/plot_scripts/plots/fig04.png}

  \caption{Bloom timing (bloom start, peak, and end day) for each sea area along
 the routes in Figure \ref{fig:transect}, averaged over the period 2000 to 2014,
 and for all applied bloom metrics. Whiskers indicate standard deviations over t
he 15-year study period. The bloom peak-day is independent of the chosen me
tric,metric andthus plotted separately. The sea areas are ord
ered by latitude, from south to north. }
\label{fig:phenology_geo_timing}
\end{figure}

---

## Author Response (AR2)

**Response to reviewers on 'Spring Blooms in the Baltic Sea have weakened but lengthened from 2000 to 2014' by P. M. M. Groetsch et al.**

**Anonymous Referee #2**

- The use of the term "variable-theshold" could still lead to confusion (i.e., annual median vs. climatological median). I suggest using "spatially variable-threshold" or similar.

- The sentence: "The observed negative tren in peak concentration introduces an artificial negative trend in bloom duration" can create confusion, because it actually does not (as it is explained afterwards). I suggest changing to "was expected to introduce".

*Authors' response: We implemented both recommendation in the manuscript.*

```
diff --git a/AlgalineSpringBloom.tex b/AlgalineSpringBloom.tex
index e2e11b6..47a062b 100644
--- a/AlgalineSpringBloom.tex
+++ b/AlgalineSpringBloom.tex
@@ -114,7 +114,7 @@ Alg@line chla concentrations (see section \ref{ssec:algalin
e}) were resampled to
```

1) Chla concentration exceeding a fixed-threshold of \SI{5}{\milli\gram\per\cubic\meter} w
as defined as bloom by \citet{Fleming2006}, further referred to as \texttt{const5}. A 21-d
ay centred-running-mean filter was used to keep results comparable to the other metrics co
nsidered, whereas \citet{Fleming2006} used a 7-day centred-running-median filter.

2) \citet{Siegel2002} proposed a spatially variable-threshold metric based on the
5 \%-above-median concentration, but reported small quantitative differences for threshold
s between 1 and 30 \%-above-median. Their threshold is based on the complete annual cycle,
 while here only the spring bloom period from day-of-year 31 to 160 is considered. We refe
r to this metric as \texttt{median5}.

3) Distributions proposed to describe bloom phenology include shifted-Gaussian \citep{Plat
t2009}, Gamma \citep{Vargas2009}, and Weibull distributions \citep{Rolinski2007}. The shif
ted Gaussian is symmetric in shape, whereas Gamma distributions allow for different slopes
 of bloom rise and decline. In addition, Weibull functions recognize non-zero offsets befo
re and after the bloom phase. The latter has proven essential to obtain a good fit for the
 transition phase between spring and summer bloom with the here analysed data set. A modif
ied Weibull-function, as proposed by \cite{Rolinski2007}, was fitted non-linearly to the p
reprocessed and scaled (to a range of zero to one) chla concentrations. The bloom initiati
on and end are defined as the $10^{th}$ and $90^{th}$ percentiles before and after the blo
om peak, respectively. This metric is further referred to as \texttt{weibull}.

@@ -137,7 +137,7 @@ Determination of the fluorescence yield was supported by an
 'adaptive regression
Figure \ref{fig:diurnal_variability}A shows normalized fluorescence observations as a func
tion of sampling time-of-day. Results are presented separately for summer (May to August),
 winter (November to February) and the transition periods (autumn, spring). Diurnal variab
ility was most pronounced in summer, when the fluorescence signal varied on average 50 \%
over the course of a day. In winter and during the transition periods (spring, autumn) a d
iurnal variability of 35 and 38 \%, respectively, was contained in the fluorescence signal
s. This seasonal effect is likely caused by variations in average irradiance intensity, wh
ich are modulated primarily by sun elevation, but also by atmospheric conditions (e.g. clo
ud cover, aerosol optical thickness) and optical properties of the water body (e.g. ice co
ver, attenuation). Figure \ref{fig:diurnal_variability}B depicts normalized fluorescence a
s a function of solar elevation. In this representation seasonal differences in diurnal va
riability are essentially absent and the correspondence between solar elevation and averag
e fluorescence response was approximately linear for daytime observations.

\subsection{Bloom Intensity and Timing}
Blooms generally developed first in the south and progressed towards the north (see Fig. \
ref{fig:phenology_geo_timing} and Table \ref{tab:bloomstats}). Bloom peak timing (not infl
uenced by choice of metric) followed this pattern, as did metric-dependent bloom start and
 end dates. The fixed-threshold bloom metric \texttt{const5} suggested longer blooms in hi
gh-biomass sea areas like the \textsc{gof}, compared to low-biomass areas such as the \tex
tsc{sbs}. The spatially variable-threshold metric \texttt{median5} applies area-sp
ecific bloom thresholds (\textsc{nbp}: \SI{3.52}{\milli\gram\per\cubic\meter}, \textsc{gof
}: \SI{4.95}{\milli\gram\per\cubic\meter}, \textsc{got}: \SI{2.51}{\milli\gram\per\cubic\m
eter}, \textsc{sbs}: \SI{2.62}{\milli\gram\per\cubic\meter}, \textsc{bom}: \SI{4.02}{\mill
i\gram\per\cubic\meter}) and resulted in approximately stable bloom duration in all sea ar
eas. The \texttt{weibull} metric, which is not sensitive to absolute bloom intensity, also
 resulted in comparable bloom durations for all sea areas. The year-to-year variability of
 start, peak, and end days generally increased towards the south for all metrics.

Spring bloom intensity was described by three parameters: the metric-independent bloom pea
```

k concentration (\textsc{peakheight}), the chla concentration average during bloom conditions (\textsc{concavg}), and the sum of daily chla concentrations over the bloom period (\textsc{bloomidx}). Similar patterns were observed for all these parameters and bloom metrics, as illustrated in Fig. \ref{fig:phenology_geo_intensity}. The highest bloom intensity was found in the \textsc{gof} and \textsc{nbp}, followed by the \textsc{bom}. Low-intensity blooms were observed in the \textsc{sbp} and the \textsc{got}. Variability was generally proportional to bloom intensity, highest in the high-biomass and coastal \textsc{gof} and \textsc{bom}. Variability in \textsc{bloomidx} was comparable to that in \textsc{peakheight}, while \textsc{concavg} was considerably more stable. All calculated bloom phenology parameters can be found in the supplementary material.

@@ -183,7 +183,7 @@ Interannual variability in coastal systems exceeds long-term trends by orders of

\cite{Helcom2014} reported stable or increasing chla concentrations for the period 2007-2011 in several Baltic Sea areas despite signs of declining nutrient concentrations. More recently, eutrophication trend reversal and oligotrophication processes were reported by \cite{Andersen2015}, based on analysis of 112 years of consolidated Baltic Sea observations. Both reports considered surface-layer chla concentration in summer as one of the direct indicators for eutrophication, but did not include spring bloom in their assessment. The time series for 2000-2014 that we present here fills this gap: a negative trend in bloom intensity was also found for spring bloom, providing further evidence for their hypothesis of gradual nutrient load reduction.

Thresholds of \texttt{const5} and \texttt{median5} are fixed for the whole time series. The observed negative trend in peak concentration introduceswas expected to introduce an artificial negative trend in bloom duration because an increasingly higher percentile of the distribution is seen below the bloom threshold (Fig. \ref{fig:trend_scheme}). Contrary to this expected behaviour, however, \texttt{const5} and \texttt{median5} revealed no significant trends in bloom duration. This indicates that the anticipated negative trend in bloom duration was countered by a positive trend, e.g. in bloom intensity. The Weibull-metric is based on concentration distribution-ratios that are calculated individually for each bloom. Therefore, Weibull-metric results for bloom duration are not sensitive to long-term trends in peak concentration. Weibull-distribution metrics confirmed a highly significant, positive trend in bloom duration. These two sets of results corroborate the conclusion that spring blooms in the Baltic Sea have become longer, while chla peak and average concentration levels declined.

This 'flattening' of the concentration distribution is supported by the absence of a trend in time-integrated biomass \textsc{bloomidx} and by shifts in nutrient concentration timing (earlier nutrient peak concentration, later 25 \%-of-peak-value day). These results indicate that annually generated spring bloom biomass has not changed significantly over the study period, in contrast to bloom timing. \cite{Kahru2014} found a similar development for cyanobacterial summer surface bloom, and reported decadal oscillations, yet no long-term trend, of surface area covered by cyanobacteria in the period 1979-2013. In the same period, summer bloom initiation moved to earlier dates by \SI{-0.6}{\day\per\year}. These results suggest that the gap has decreased between dinoflagelate- and diatom-dominated spring bloom and cyanobacterial summer bloom. Due to the shorter period covered here as compared to the time series presented by \cite{Kahru2014}, it cannot be ruled out that the spring bloom trends are caused by decadal oscillation. Moreover, Alg@line nutrient records often did not commence sufficiently early in the season to record bloom onset. Trends in bloom start and nutrient peak timing can therefore not be derived at the same accuracy and precision as the other phenological parameters. In future, additional data and longer time series may revise this analysis. To this end, nutrient metrics derived in this work are provided in the appendix.